# Methamphetamine induces cardiomyopathy by Sigmar1 inhibition-dependent impairment of mitochondrial dynamics and function

Chowdhury S. Abdullah [1,9], Richa Aishwarya[2,9], Shafiul Alam [1], Mahboob Morshed[1], Naznin Sultana Remex[2], Sadia Nitu[1], Gopi K. Kolluru[1], James Traylor[1], Sumitra Miriyala [3], Manikandan Panchatcharam [3], Brandon Hartman[1], Judy King[1], Mohammad Alfrad Nobel Bhuiyan[4], Sunitha Chandran[5], Matthew D. Woolard[5], Xiuping Yu[6], Nicholas E. Goeders[7], Paari Dominic[8], Connie L. Arnold[8], Karen Stokes[2], Christopher G. Kevil [1,2,3], A. Wayne Orr[1,2,3] & Md. Shenuarin Bhuiyan [1,2✉]

Methamphetamine-associated cardiomyopathy is the leading cause of death linked with illicit drug use. Here we show that Sigmar1 is a therapeutic target for methamphetamine-associated cardiomyopathy and defined the molecular mechanisms using autopsy samples of human hearts, and a mouse model of "binge and crash" methamphetamine administration. Sigmar1 expression is significantly decreased in the hearts of human methamphetamine users and those of "binge and crash" methamphetamine-treated mice. The hearts of methamphetamine users also show signs of cardiomyopathy, including cellular injury, fibrosis, and enlargement of the heart. In addition, mice expose to "binge and crash" methamphetamine develop cardiac hypertrophy, fibrotic remodeling, and mitochondrial dysfunction leading to contractile dysfunction. Methamphetamine treatment inhibits Sigmar1, resulting in inactivation of the cAMP response element-binding protein (CREB), decreased expression of mitochondrial fission 1 protein (FIS1), and ultimately alteration of mitochondrial dynamics and function. Therefore, Sigmar1 is a viable therapeutic agent for protection against methamphetamine-associated cardiomyopathy.

[1] Department of Pathology and Translational Pathobiology, Louisiana State University Health Sciences Center-Shreveport, Shreveport, LA 71103, USA. [2] Department of Molecular and Cellular Physiology, Louisiana State University Health Sciences Center-Shreveport, Shreveport, LA 71103, USA. [3] Department of Cell Biology and Anatomy, Louisiana State University Health Sciences Center-Shreveport, Shreveport, LA 71103, USA. [4] Division of Biostatistics and Epidemiology, Cincinnati Children's Hospital, Cincinnati, OH 45229, USA. [5] Department of Microbiology and Immunology, Louisiana State University Health Sciences Center-Shreveport, Shreveport, LA 71103, USA. [6] Department of Biochemistry and Molecular Biology, Louisiana State University Health Sciences Center-Shreveport, Shreveport, LA 71103, USA. [7] Department of Pharmacology, Toxicology and Neuroscience, Louisiana State University Health Sciences Center-Shreveport, Shreveport, LA 71103, USA. [8] Department of Medicine, Louisiana State University Health Sciences Center-Shreveport, Shreveport, LA 71103, USA. [9]These authors contributed equally: Chowdhury S. Abdullah, Richa Aishwarya. ✉email: mbhuiy@lsuhsc.edu

Methamphetamine (METH) is a highly addictive ampheta-mine class psychostimulant drug whose illicit use is dra-matically increasing in the United States[1,2]. It has been reported that ~1.6 million Americans used METH in 2017, with 964,000 cases (up from 684,000 cases in 2016) of emergency room visits owing to METH abuse associated disorders[3–5]. METH is toxic to both neuronal and cardiovascular systems; cardiovascular disease is the second leading cause of death among METH abusers following accidental overdose[6,7]. METH abuse associated with car-diovascular toxicity includes hypertension, tachycardia, cardiac arrhythmia, acute coronary vasospasm, myocardial infarction, and cardiomyopathy[1,2,6–8]. Clinical case studies of the hearts of human METH users have revealed severely reduced left ventricular (LV) ejection fraction, LV chamber dilatation, myocardial lesions, cardiac hypertrophy, fibrosis, and inflammation in relatively younger patients (≤50 years) with no other significant cardiac risk factors[6,9–11]. Unfortunately, reports of the histopathological characteristics present in the hearts of METH users are scarce and most of the studies used only a very small number of samples[9,12,13].

METH increases dopamine levels in dopaminergic neuronal synapses by competitive inhibition of dopamine uptake from synapses, and it stimulates dopamine efflux from neuronal vesicular stores[14]. The presence of excess dopamine in the extracellular spaces rewires the brain reward circuit promoting repetition of the pleasurable activity after taking the drug[14,15]. Despite a long pharmacological half-life ($t_{1/2} = 12$ hours), an immediate METH-induced intense "rush" or "flash," described as an extremely plea-surable experience, lasts only a few minutes following the injection and dissipates before the drug concentration in the blood has fallen significantly. Hence, to maintain the "high," METH users indulge in a risky behavior known as "binge and crash," where users continue to take METH for several days (typically for 4 days in a week, known as "run") while foregoing food and sleep, followed by a period of abstinence[16,17]. METH's "binge and crash" use pattern is a common behavioral trait in its human users, along with chronic use patterns. Importantly, binge users tend to take ~1.5 times more METH per month than chronic users, running the risk of overdose[16,17]. Acute and chronic METH treatment in rodents showed the evidence of METH-induced cardiotoxicity[18–20]. Find-ings from these studies suggest that METH causes mitochondrial dysfunction, increases oxidative stress, alters intracellular $Ca^{2+}$ dynamics, enhances inflammatory marker, and adversely affects cardiac contractility[20–22]. However, the molecular targets and mechanisms of METH-induced cardiomyopathy remain elusive.

Sigma-1 receptor (Sigmar1) is a widely expressed intra-organelle signaling modulator that interacts physically with client proteins to regulate cellular calcium homeostasis[23], endoplasmic reticulum (ER) stress, cell survival[24], and to reg-ulate gene transcription[25]. Extensive Sigmar1 ligand-based studies indicate that activation (expression) of Sigmar1 con-fers neuroprotection in neurodegenerative diseases[26,27]. Inter-estingly, Sigmar1 has been found to be a direct target of METH as METH binds to Sigmar1 at physiological concentrations ($K_i$ value of ~2 μM)[28]. Recently, activation of Sigmar1 using the Sigmar1 agonist PRE-084 has been shown to limit METH-induced dopamine efflux in dopaminergic neurons, attenuate METH-induced locomotion, and motivate behavior and the drug-seeking brain reward function[29]. We have previously reported that Sigmar1 is highly expressed in the heart[30,31], and that the genetic loss of Sigmar1 causes altered cardiac mito-chondrial morphology and dysfunction, leading to cardiac contractile dysfunction[32]. Despite all these studies suggesting Sigmar1's potential role in cardiac physiology and also serve as a direct target of METH, to our knowledge, no studies to date explored the direct involvement of Sigmar1 in the development of METH-associated cardiac pathology.

METH-induced cardiomyopathy is a poorly characterized disease entity as METH-induced molecular perturbations and his-topathological changes in the human heart remain poorly char-acterized. We investigated the pathological remodeling of the heart associated with illicit METH use with tissue obtained from the autopsy of the hearts of human METH users and non-METH users based on toxicological reports. We also used a preclinical "binge and crash" METH administration mouse model to recapitulate the pathological features observed in the hearts of human METH users and determine the molecular mechanisms of METH-induced car-diomyopathy. We performed histological, gravimetric, and echo-cardiographic studies of cardiac tissue, mitochondrial bioenergetics assays, assessments of mitochondrial dynamics regulatory protein signaling, performed high-resolution mitochondrial morphometry measurements in the hearts of METH-treated mice, and immu-nohistochemistry analyses in human autopsy hearts to define the molecular targets of METH cardiomyopathy.

## Results

### The hearts of human METH users demonstrate adverse fibrotic remodeling regardless of the cause of death.

Accumulating evidence indicates that METH use is associated with the devel-opment of cardiomyopathy[9–13,33]. However, a majority of the literature is limited to cursory case reports of the clinical data of one or several patients without providing any insight into the common histopathological alterations, if any, present in the hearts of human METH users. In this study, we examined the tissue from 34 humans LV heart sections from autopsy (patient demographics and physiological data in Supplementary Table 1) who were determined to be METH positive as the result of tox-icological studies. Among the heart samples from METH users, six patients died owing to METH overdose-related toxicity (METH toxicity group), whereas the cause of death of the remaining patients was from violence, accidents, combined drug overdose, or was acute myocardial infarction related (Supple-mentary Table 1). We also collected autopsy tissue from seven human hearts (non-METH user group) from patients whose deaths were accidental (Supplementary Table 1) and who were negative for METH and other drugs following toxicological stu-dies. The LV sections from non-METH and METH users were stained with Picro Sirius Red (PSR) to detect collagen deposition in the myocardium (Fig. 1). The heart tissue from both METH toxicity and METH users exhibited increased collagen deposition (stained red in PSR staining) in the perivascular and interstitial spaces of the myocardium compared with heart tissue from non-METH users (Fig. 1a). The collagen content (percent collagen area) was significantly increased in the hearts of METH users compared with those of non-METH users (Fig. 1b) after adjust-ment for age (Fig. 1d), sex, race, heart weight (Fig. 1c), body weight, height, and BMI (Fig. 1e). We carried out Masson's Trichrome staining and found a similarly consistent pattern of fibrotic remodeling in the perivascular and interstitial spaces in the myocardium of METH users compared with that of non-METH users (Fig. 2a). Fibrotic areas (percent fibrosis area) were significantly increased in the LV sections of METH users com-pared with those of non-METH users after correction for vari-ables including age, sex, race, heart weight, body weight, height, and BMI (Fig. 2b). Overall, we found that the hearts of human METH users exhibit fibrotic remodeling, irrespective of the cause of death, underscoring the underappreciated cardiotoxic pathol-ogies associated with METH abuse in humans.

### "Binge and crash" METH exposure in mice induces cardiac hypertrophy and adverse fibrotic remodeling with cardiac func-tional decline.

To study the cellular and molecular mechanisms of

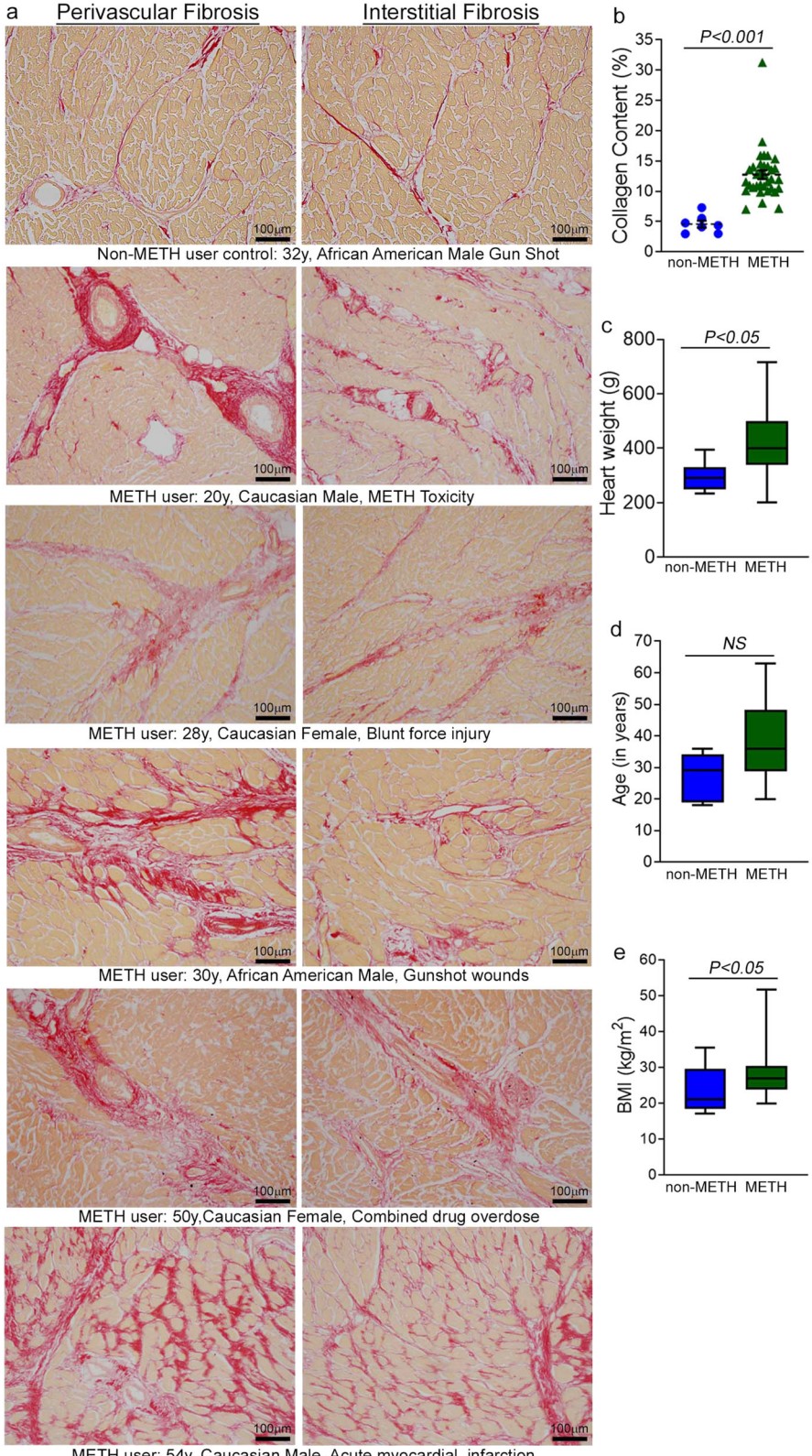

**a** Perivascular Fibrosis  Interstitial Fibrosis

Non-METH user control: 32y, African American Male Gun Shot

METH user: 20y, Caucasian Male, METH Toxicity

METH user: 28y, Caucasian Female, Blunt force injury

METH user: 30y, African American Male, Gunshot wounds

METH user: 50y,Caucasian Female, Combined drug overdose

METH user: 54y, Caucasian Male, Acute myocardial infarction

METH-induced direct cardiotoxicity and corroborate our findings in the hearts of human METH users, we used a "binge and crash" METH administration protocol in mice. This preclinical "binge and crash" METH administration protocol mimics the conventional METH "binge and crash" use pattern observed in humans[34]. Following 4 weeks of "binge and crash" METH administration, we assessed the cardiac dimensions and function using M-mode echocardiography (Fig. 3)[32,35–38]. Cardiac systolic parameters, i.e., percent fractional shortening (%FS) (Fig. 3c) and percent ejection fraction (Fig. 3f) were significantly reduced with increased LV internal dimension at systole (LVID;s) (Fig. 3a) and increased LV volume at systole (LV Vol;s) (Fig. 3d) in METH-treated mice compared with vehicle-treated mice. There were no significant changes in LV internal dimension at diastole (LVID;d) (Fig. 3b), LV

**Fig. 1 Increased collagen deposition in the myocardium of human METH-user heart autopsy tissue. a** Picro Sirius Red (PSR) staining was conducted in non-METH and METH user human LV heart sections to assess the extent of collagen deposition. Formalin-fixed, paraffin-embedded LV sections (5 μm) from non-METH ($n = 7$), and METH users ($n = 34$) were stained with PSR. The METH user group consists of heart samples from patients who died from acute METH intoxication/overdose and patients positive for METH following toxicology studies carried out in the case of accidental causes of death. Representative images shown in the left panel images demonstrate collagen deposition (in red) in the perivascular region. Those in the right panel images demonstrate the extent of collagen deposition (in red) in the interstitial spaces in the same non-METH and METH users heart tissue. Scale bar: 100 μm. **b** Dot plot represents individual values of quantified percent collagen content in PSR-stained LV sections from the non-METH ($n = 7$) and METH ($n = 34$) user groups. The collagen content (red-stained area) relative to the total area of the myocardium was measured in 10–15 high-magnification microscopic fields (×10) for each heart section and averaged for analysis. $P$ values were determined between non-METH and METH groups by constructing multiple linear regression analysis with models adjusted for age, sex, race, heart weight, body weight, height, and body mass index. Data are expressed as mean ± SEM. A $P < 0.05$ between groups was considered statistically significant. **c–e** Heart weight **c**, age **d**, and body mass index (BMI) **e** data were tested for normality by analysis using the Kolmogorov–Smirnov test followed by the Kruskal–Wallis test for $P$ value determination. Boxes depict interquartile ranges, lines represent medians, and whiskers represent ranges. $P < 0.05$ between groups was considered statistically significant. *NS* not significant.

volume at diastole (LV Vol;d) (Fig. 3e), interventricular septum thickness at systole and diastole (IVS;s and IVS;d) (Fig. 3h, i), or LV posterior wall thickness at systole and diastole (LVPW;s and LVPW;d) (Fig. 3j, k). Heart rates were similar between vehicle- and METH-treated mice during echocardiography recordings (Fig. 3g). In summary, "binge and crash" METH administration caused a systolic cardiac functional decline in mice.

Next, we assessed the gross and microscopic heart morphometry to evaluate METH treatment-induced changes in the macro- and microscopic structures of the mouse heart. The hearts from METH-treated mice were visually larger (Fig. 4a, left panel), evidence for which was also demonstrated by gravimetric analyses showing increased heart weight (mg)-to-tibia length (cm) ratio in the hearts of METH-treated mice (Fig. 4a, right panel) compared with those of vehicle-treated mice. To confirm whether treatment with METH-induced hypertrophy in cardiomyocytes, we stained the hearts of vehicle- and METH-treated mice with wheat germ agglutinin (WGA). We found increased cross-sectional areas (μm$^2$) in the cardiomyocytes from METH-treated mice compared with those of the vehicle-treated group in WGA-stained heart sections (Fig. 4b). mRNA expression analysis revealed activation of molecular profiles of cardiac hypertrophy with increased fetal cardiac gene expression, i.e., *Nppa*, *Nppb*, *Myh7*, and decreased *Myh6* expression (Fig. 4c). However, "binge and crash" METH administration did not significantly affect the body weights of the mice (Fig. 4d). Heart tissues were stained with Masson's Trichrome (Fig. 5a, left panel) and PSR (Fig. 5b, left panel) to evaluate fibrosis and collagen deposition in the myocardium following 4 weeks of "binge and crash" METH administration. Our image analysis revealed significant upregulation of the percent fibrosis area (Fig. 5a, right panel) and percent collagen deposition (Fig. 5b, right panel) in the METH-treated group compared to the vehicle-treated group. Immunoblot analyses of protein markers of fibrosis, periostin, and α smooth muscle actin confirmed the activation of fibrotic signaling in the hearts of METH-treated mice (Fig. 5c).

To evaluate the effects of METH on the immune cell population, we carried out immune phenotyping in the blood and splenocytes of vehicle- and METH-treated mice using flow cytometry. We used a panel of antibodies to assess the B cell, T cell, NK cell, polymorphonuclear cell, and monocyte/macrophage populations in the blood and splenocytes. We did not find any significant change in the percentage of the immune cell population in the blood and splenocytes (Supplementary Fig. 1a, b). Analysis of monocyte/macrophage populations demonstrated a decrease in the percentage of Ly6C$^+$ classical pro-inflammatory macrophages and a slightly increased percentage of Cx3cr1$^+$Ly6c$^-$ macrophages in the spleen. In contrast, both populations were unchanged in the blood (Supplementary Fig. 1c, d). Therefore, "binge and crash" METH administration did not alter selective innate and adaptive immune cell populations in the spleen or systemic circulation.

**METH causes attenuation of mitochondrial fission regulatory protein Fis1 expression in the heart.** Mitochondrial dynamics are orchestrated through a combination of mitochondrial fusion and fission events. It has been well documented that an imbalance between fusion and fission events affects heart mitochondrial function, resulting in heart failure[39–41]. We determined whether METH affects cardiac mitochondrial dynamics regulatory protein expression. Our immunoblot analyses in whole-cell lysates and isolated mitochondrial fractions revealed that METH causes significant downregulation of mitochondrial fission regulatory protein Fis1 (mitochondrial fission 1 protein, Fis1) level in whole-cell (Fig. 6a) fractions reflecting in reduced Fis1 localization in the mitochondrial fraction (Fig. 6b) in the heart. However, we did not find any differences in the protein levels in mitochondrial fission regulatory proteins, i.e., Drp1 (dynamin-related protein 1) and Mff (mitochondrial fission factor), or in mitochondrial fusion regulatory proteins, i.e., OPA1 (Optic Atrophy 1), Mfn2 (mitofusin 2), and OPA1 processing metalloprotease OMA1 (overlapping with the M-AAA protease 1 homolog) in whole-cell lysate and mitochondrial fractions (Fig. 6a, b). We probed GAPDH on the same membrane of whole-cell and mitochondrial fractions to assess the purity of isolated mitochondrial fractions (Fig. 6a, b). We also used mitochondrial outer membrane integral protein, Tom20, and mitochondrial inner membrane integral protein, Tim23, to confirm the mitochondrial extracts in vehicle- and METH-treated mouse hearts (Fig. 6b). We found that METH selectively attenuated mitochondrial fission regulatory protein Fis1 level in the heart.

**METH exposure causes a hyperfused mitochondrial network that coincides with suppression of mitochondrial respiration in cardiomyocytes.** To evaluate whether METH-induced downregulation of Fis1 in the heart has any functional consequences in the mitochondrial morphology in myocytes, we conducted transmission electron microscopy studies in the hearts of vehicle- and METH-treated mice (Fig. 7a–e). Remarkably, "binge and crash" METH treatment for 4 weeks caused the appearance of abnormally shaped, large, hyperfused mitochondria (Fig. 7b) as evidenced by the decreased number of mitochondria per microscopic field (Fig. 7c) and increased average mitochondrial areas (Fig. 7d) with an increased percentage of larger (≥0.5 μm$^2$) mitochondria (Fig. 7e) compared with observations in the hearts of vehicle-treated mice. Systemic administration of METH to mice may exert METH-induced excess catecholamine-mediated indirect effects on the mitochondrial structure and respiration in the hearts of METH-treated mice in vivo (Figs. 6, 7a–e). To

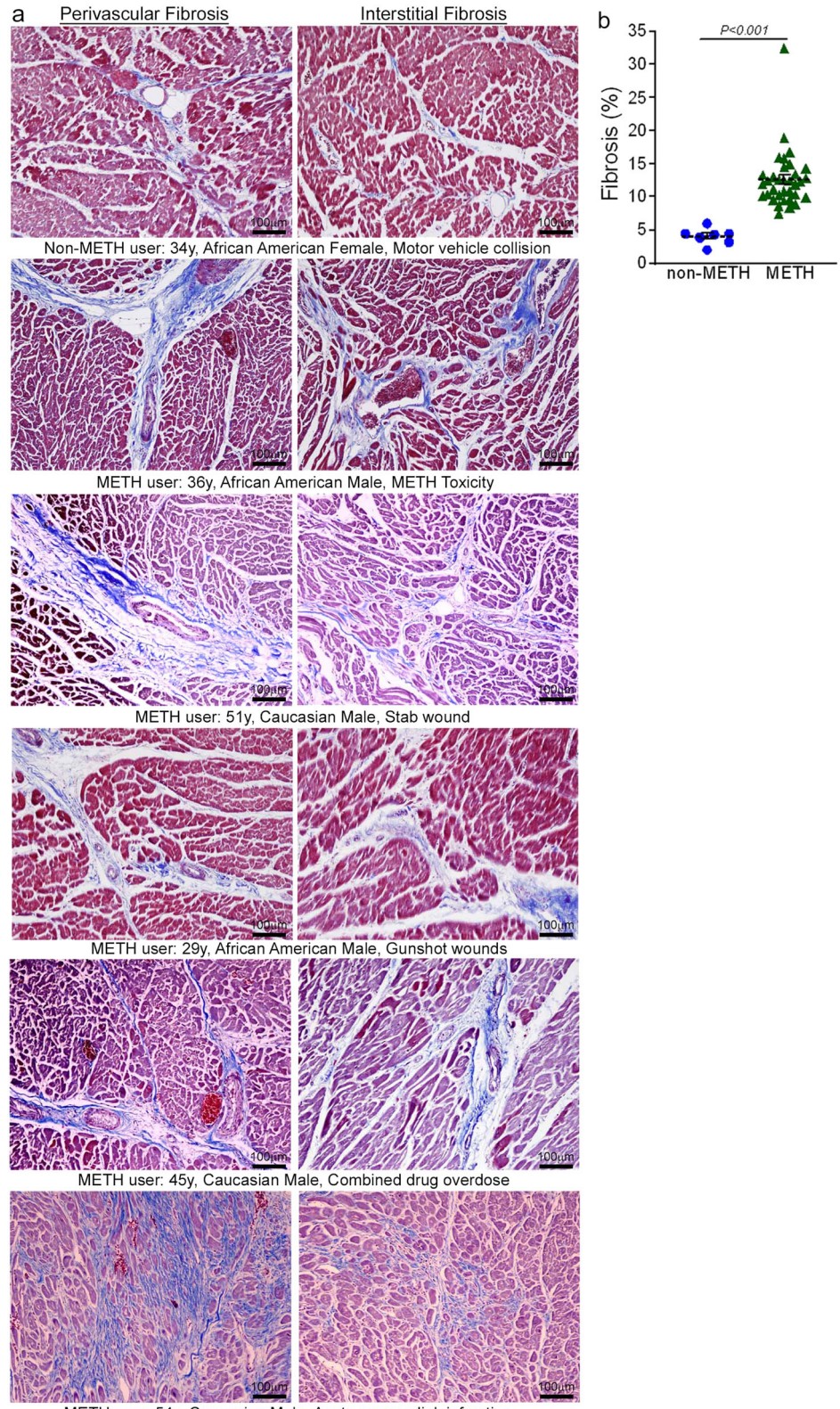

obviate the METH-induced neurohumoral impact on the heart and to study the direct METH exposure-associated alteration in mitochondrial morphology and respiration of cardiomyocytes, we exposed isolated primary neonatal rat cardiomyocytes (NRC) to METH for 24 hours. Notably, exposure of NRC to METH (100 μM) for 24 hours resulted in a hyperfused, extended tubular mitochondrial network compared to the short tubular mitochondrial system observed in vehicle-treated cardiomyocytes following visualization with MitoTracker Red staining (Fig. 7f). Measurement of the mitochondrial network length showed a significant increase in mitochondrial length (Fig. 7g) and mitochondrial aspect ratio (ration of mitochondrial length-to-diameter) (Fig. 7h), with an increased long mitochondrial percentage in the 4 μm to >20 μm range as shown in the

**Fig. 2 Enhanced fibrotic remodeling in the myocardium of human METH-user heart autopsy tissues. a** Masson's Trichrome staining was conducted in non-METH and METH user LV sections to assess the interstitial and perivascular fibrosis in the myocardium. Formalin-fixed, paraffin-embedded LV autopsy sections (5 µm) from non-METH ($n = 7$), and METH users ($n = 34$) were used for Masson's Trichrome staining. The METH user group consists of heart samples from patients who died from acute METH intoxication/overdose and patients who were positive for METH following toxicology studies carried out in the case of accidental, combined drug overdose, or acute myocardial infarction-related causes of death. Representative left panel images to demonstrate the extent of fibrosis (in blue) in the LV perivascular region, and right panel images show the fibrosis (in blue) in the interstitial space in the same non-METH and METH user hearts. Scale bar: 100 µm. **b** Dot plot represents the individual values of the quantified percent fibrosis area in Masson's Trichrome stained LV sections in non-METH ($n = 7$) and METH ($n = 34$) user groups. The size of the fibrosis area (blue-stained area) relative to the total myocardium area was measured in 10–15 high-magnification microscopic fields (×10) for each heart section and averaged for analysis. $P$ values were determined between non-METH and METH groups by constructing multiple linear regression analysis with models adjusted for age, sex, race, heart weight, body weight, height, and body mass index. $P < 0.05$ between groups was considered statistically significant. Data are expressed as mean ± SEM.

mitochondrial length distribution frequency graph (Fig. 7i) in METH-treated cardiomyocytes compared with those treated with vehicle. We found that treatment with METH directly alters mitochondrial network morphology in cardiomyocytes, confirming our observations of METH-induced impaired mitochondrial fission signaling (Fig. 6) and morphometry (Fig. 7).

**METH causes suppression of mitochondrial respiration in mouse heart.** As METH administration causes altered mitochondrial dynamics and morphometry, we investigated the effects of METH treatment on cardiac mitochondrial respiration after "binge and crash" METH treatment. Our real-time oxygen consumption rate (OCR) measurement studies revealed reduced basal OCR, the sum of all physiological mitochondrial oxygen consumption, in an isolated mitochondria preparation from the hearts of METH-treated mice compared to those of the vehicle-treated group (Fig. 8a, b). Inhibition of mitochondrial ATP synthase (Complex V) by oligomycin causes a drop in basal OCR that allows ATP-linked OCR to be calculated (Fig. 8a). Although METH treatment attenuated basal mitochondrial respiration, ATP-linked respiration was unchanged (Fig. 8c) compared to that in vehicle-treated heart mitochondria. Next, carbonyl cyanide 4-(trifluoromethoxy) phenylhydrazone (FCCP) was injected to uncouple respiration from oxidative phosphorylation to measure maximal OCR (Fig. 8d). METH-treated heart mitochondria exhibited a significantly lower maximal OCR when compared with vehicle-treated heart mitochondria (Fig. 8d). To quantify the non-mitochondrial OCR, rotenone, and antimycin A was added to inhibit mitochondrial Complex I and Complex III, respectively (Fig. 8a). The non-mitochondrial related OCR was similar between vehicle- and METH-treated heart mitochondria (Fig. 8e). The mitochondrial reserve respiratory capacity, calculated by subtracting the basal OCR from the FCCP-stimulated OCR, was markedly lower in METH-treated heart mitochondria compared with those of the vehicle-treated group (Fig. 8f). Also, ATP turnover (Fig. 8g), calculated by subtraction of ATP-linked OCR from basal OCR, and maximal respiration (Fig. 8h), calculated by subtraction of non-mitochondrial OCR from FCCP-stimulated OCR, were significantly lower in METH-treated heart mitochondria compared with those in the vehicle-treated group. To monitor whether METH-mediated mitochondrial dysfunction is progressive, we also observed the mitochondrial respiration after 2 weeks of "binge and crash" METH treatment (Supplementary Fig. 2a–h). We also observed significantly altered mitochondrial respiratory parameters in the METH-treated hearts indicating defects in mitochondrial respiration occur in an early stage of METH treatment (Supplementary Fig. 2a–h). In summary, we found that "binge and crash" METH administration suppressed mitochondrial oxidative electron transport chain-dependent respiration in cardiac mitochondria.

Next, we sought to assess METH-induced mitochondrial oxidative respiration in isolated neonatal cardiomyocytes (NRC)

to test the direct effect of METH on mitochondrial respiration. As the mitochondrial isolation procedures used in the pathologically remodeled heart may compromise mitochondrial integrity during the isolation process, we measured mitochondrial respiration in intact NRC seeded in Seahorse analyzer plates treated with vehicle and METH (100 µM) for 24 hours. Notably, METH treatment of cardiomyocytes suppressed basal OCR (Supplementary Fig. 3a, b) and FCCP-stimulated OCR (Supplementary Fig. 3a, d) with no change in ATP-linked OCR following oligomycin injection (Supplementary Fig. 3a, c) or non-mitochondrial OCR following rotenone and antimycin injection (Supplementary Fig. 3a, e) compared with results observed in vehicle-treated cardiomyocytes. METH exposure also significantly lowered the mitochondrial bioenergetics parameters, including reserve capacity (Supplementary Fig. 3f), ATP turnover (Supplementary Fig. 3g), and maximal respiration (Supplementary Fig. 3h) compared with those in the vehicle-treated group. To rule out the possible involvement of METH-induced catecholamine-mediated indirect effects on the mitochondrial respiration, we also treated the NRC with norepinephrine (NE) (5, 25, or 50 µM) for 24 hours (Supplementary Fig. 4a–h). Treatment of cardiomyocytes with NE did not significantly change the key OCR parameters (Supplementary Fig. 4a–h), suggesting that the mitochondrial dysfunction caused by METH treatment represents a direct effect of METH on cardiomyocytes. In summary, we found that METH caused a defective mitochondrial network morphology that coincided with compromised mitochondrial respiration in cardiomyocytes.

**METH downregulated the Fis1 level via CREB expression and activation in the heart.** Recently, the cAMP response element-binding protein (CREB) has been identified as a transcriptional regulator of the Fis1 protein levels and mitochondrial morphology in cardiomyocytes[42]. As METH reduced Fis1 protein expression in mouse hearts (Fig. 7a, b), we monitored the protein level and activation of CREB in human and mouse hearts. Our immunohistochemical (IHC) analysis revealed that METH substantially reduced phosphorylated CREB at serine133 (Ser133) in the hearts of human METH users (Fig. 9a) and those of mice exposed to "binge and crash" METH (Fig. 9b), suggesting that chronic METH attenuates CREB activation in the heart. Similarly, western blot analysis of mouse heart lysates showed significant downregulation of total CREB and phosphorylated CREB at Ser133 in the hearts of "binge and crash" METH-treated mice (Fig. 9c). In addition, immunocytochemical analysis of NRCs treated with METH (100 µM) also showed decreased nuclear localization of pCREBSer133 (Fig. 9d). We conclude that METH directly attenuates CREB protein level and activation through an undefined mechanism in cardiomyocytes, which might underlie METH-induced downregulation of the mitochondrial fission regulatory protein Fis1 level in cardiomyocytes.

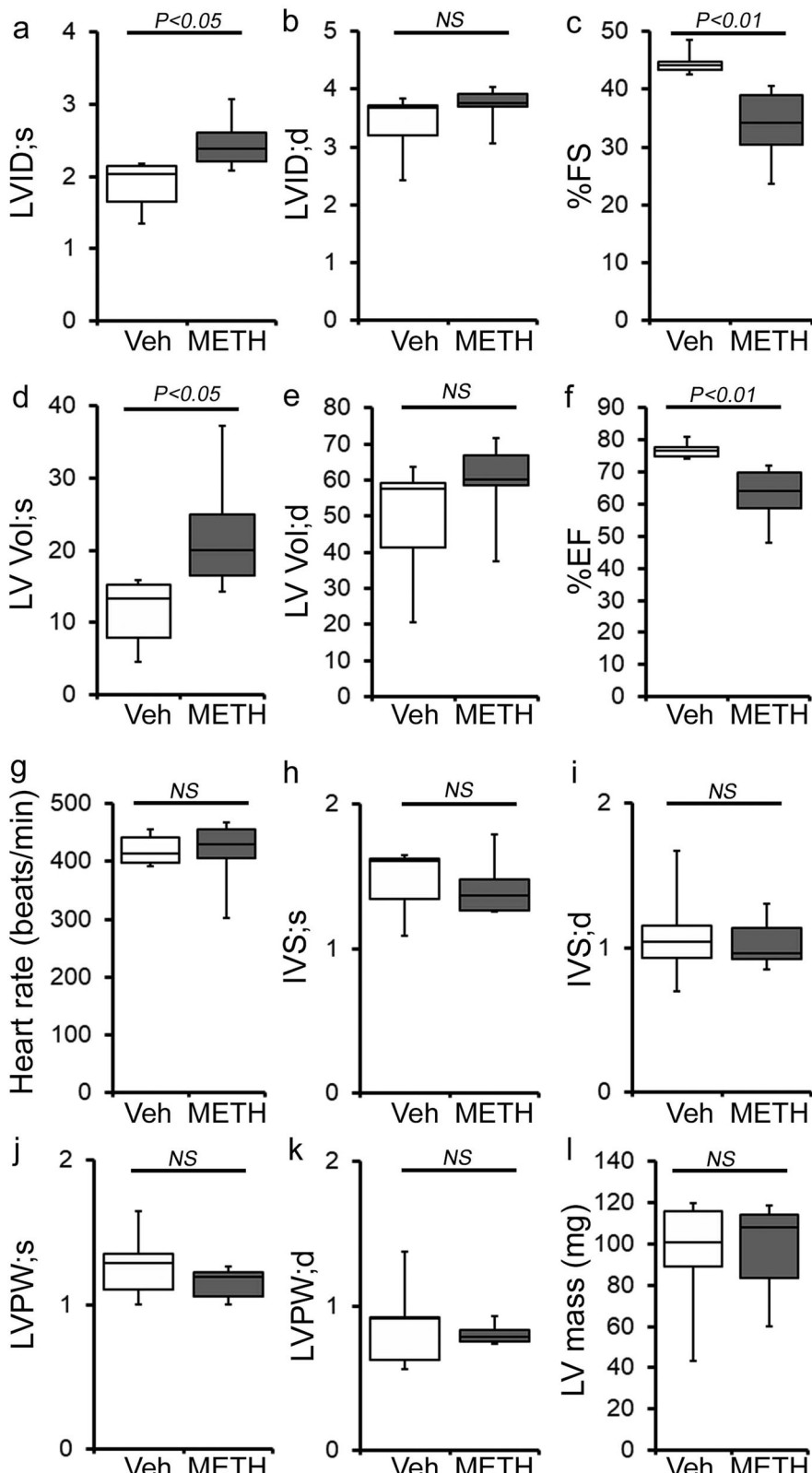

**METH-induced inhibition of Sigmar1 underlies CREB/pCREB-Ser133/Fis1 axis attenuation in cardiomyocytes**. Sigmar1 has been found to be a direct target of METH as it binds to Sigmar1 at physiological concentrations ($K_i$ value of $\sim2\,\mu M$)[28]. We recently reported that genetic loss of Sigmar1 suppressed mitochondrial respiration and led to large, hyperfused mitochondria in mouse hearts[32]. Since METH suppressed mitochondrial respiration and

caused hyperfused mitochondrial morphology in cardiomyocytes, similar to our previous observations in Sigmar1 knockout hearts, we examined whether METH regulates Sigmar1 level in the heart. IHC staining of Sigmar1 revealed its diffuse localization in cardiomyocytes in the hearts of both humans (Fig. 10a, left panel) and mice (Fig. 10b, left panel). Quantitative analysis of the images showed decreased Sigmar1 level in the myocardium of human METH

**Fig. 3 METH "binge and crash" administration to mice causes cardiac functional decline. a–l** M-mode echocardiography was used to evaluate cardiac functional parameters in mice treated with vehicle (saline) (n = 5) and METH (n = 6) 4 weeks after the subcutaneous administration of vehicle or METH. METH treatment causes a significant decline in M-mode echocardiography parameters, including **c** percent fractional shortening (%FS) and **f** percent ejection fraction (%EF) compared with those in the vehicle-treated mice. **a–l** Bar graph represents **a** left ventricular (LV) systolic internal dimension (LVID; s); **b** LV diastolic internal dimension (LVID;d); **c** percent fractional shortening (%FS); **d** LV systolic volume (LV Vol;s); **e** LV diastolic volume (LV Vol;d); **f** percent ejection fraction (%EF); **g** heart rate (beats/minute); **h** LV systolic interventricular septum thickness (IVS;s); **i** LV diastolic interventricular septum thickness (IVS;d); **j** LV systolic posterior wall thickness (LVPW;s); **k** LV diastolic posterior wall thickness (LVPW;d); **l** LV mass in vehicle- (n = 5) and METH- (n = 6) treated mice. Boxes depict interquartile ranges, lines represent medians, and whiskers represent ranges. A two-tailed unpaired Student's *t* test was used to determine *P* values. *P* < 0.05 between groups was considered statistically significant. *NS* not significant, *Veh* vehicle, *METH* methamphetamine.

toxicity hearts compared to those of non-METH human hearts (Fig. 10a, right panel). Similarly, we also found decreased Sigmar1 level in the hearts of "binge and crash" METH-treated mice (Fig. 10b, right panel) compared with those of vehicle-treated mice. These studies suggest that METH treatment decreases Sigmar1 level in cardiomyocytes, thereby affecting the reduction of CREB-Fis1 signaling in cardiomyocytes.

To determine whether Sigmar1 directly regulates the CREB/pCREBSer133/Fis1 axis in cardiomyocytes, we overexpressed Sigmar1 (10 MOI) by adenoviral infection and siRNA knockdown (50 nM) in NRCs. Our western blot analyses showed that Sigmar1 overexpression upregulated CREB and pCREBSer133 levels while simultaneously increasing Fis1 in NRC (Fig. 10d). In contrast, Sigmar1 knockdown attenuated CREB and pCREBSer133 level, which coincided with reduced Fis1 level in NRC (Fig. 10d). We conclude that Sigmar1 regulated Fis1 levels through regulation of transcription factor CREB in cardiomyocytes. Hence, METH-induced decreased Sigmar1 expression underlies the alteration in CREB/pCEBSer133/Fis1 signaling that adversely affects mitochondrial dynamics, morphometry, and respiration in cardiomyocytes, providing evidence for a pathogenic molecular mechanism in METH-induced cardiomyopathy. To confirm the direct involvement of Fis1 in METH-induced altered mitochondrial dynamics and morphology, we overexpressed Fis1 in HEK293T cells and treated them with METH (Supplementary Fig. 5). Similar to cardiomyocytes, METH treatment (100 μM) in HEK293T cells significantly reduced Fis1 and CREB proteins level (Supplementary Fig. 5a). We overexpressed the N-terminal His-tagged Human Fis1 (His-hFis1) in HEK293T cells and treated them with METH (100 μM) for 24 hrs. METH treatment significantly increased the mitochondrial dynamics, and His-hFis1 overexpression significantly decreased the METH treatment-induced altered mitochondrial length and aspect ratio in HEK293T cells (Supplementary Fig. 5b). These data suggest that Fis1 plays a critical role in METH treatment-induced altered mitochondrial dynamics and morphometry.

## Discussion

METH abuse is a growing international crisis, making it the second most abused drug class following cannabis and opioids[43]. Despite reports in the literature indicating an association between METH abuse and cardiomyopathy[9–11], little is known about the histopathological features and pathological remodeling in the heart linked with the illicit use of METH. At the same time, drug overdose-related deaths are on the rise in the young and seemingly otherwise healthy users whose hearts can be used as heart transplantation donors has grown substantially[44]. In this study, we conducted microscopic histological studies on 34 human LV heart sections from autopsies of patients positive for METH use following toxicological drug screenings. Our observations revealed the consistent presence of increased collagen deposition and fibrosis in the perivascular and interstitial spaces of the LV heart sections from METH users, regardless of the cause of their

deaths, compared to LV heart sections from non-METH users. In addition, the whole heart wet weights of hearts from METH users were significantly higher than those of non-METH user hearts. Our findings confirm the presence of adverse cardiac scarring and fibrotic remodeling in the hearts of METH users that accompany the METH-induced cardiac toxicity observed in a cohort of LV heart samples from human autopsies in the US.

Human METH users exhibited plasma METH concentrations that ranged from 2 to 3 μM. In comparison, levels can reach 17 μM in individuals arrested for erratic behavior and 87 μM in postmortem samples from non-overdose patients[45,46]. Notably, a marked accumulation of an unchanged concentration of METH has been shown in METH-sensitized rat brain cortex, striatum, and heart, but not in the liver, kidney, or abdominal muscle, following a bolus injection of 4 mg/kg METH, demonstrating a viable reason for the higher toxicity METH use poses to cardiac tissue[47]. Most of the studies of METH-associated cardiomyopathy have shown evidence of systolic cardiac dysfunction, as indicated by a lower LVEF. A growing number of clinical studies have reported severe LVEF attenuation with LV dilatation in METH users indicating the inherent existence of METH-associated cardiomyopathy[6,10,33]. Studies of patients with a discharge diagnosis of cardiomyopathy and concomitant METH abuse revealed a high percentage of patients (19 of 21) with echocardiographic LV-dilation LVEF reduction[48]. A recent study of 4407 patients positive for METH use showed that 714 patients were screened for heart failure, and 450 had abnormal levels of BNP[49]. Interestingly, the prevalence of abnormal BNP in the METH-tested patient group was higher than that in either the combined METH-negative or the non-tested groups[49]. In addition, echocardiography studies showed a significant difference in normal LVEF (50–70%) and severely dysfunctional LVEF (<30%) for METH-positive patients with normal versus abnormal BNP[49]. Logistic regression analysis revealed predictors of abnormal BNP and LVEF <30% in METH-positive patients[49]. Activation of the fetal gene program (ANP, BNP, and β-MHC) in the adult heart is used as a biomarker for cardiac hypertrophy[50] and is frequently observed in the ventricles of those with severe congestive heart failure[51,52]. In our study, we used a preclinical "binge and crash" METH administration protocol in mice to model the human "binge and crash" use pattern[34]. Our longitudinal trans-thoracic echocardiography studies showed that "binge and crash" METH exposure reduces cardiac systolic functional parameters, including LVEF and LV FS. In this study, we also observed an association between a significantly increased fetal gene program (ANP, BNP, and β-MHC) and fibrotic remodeling and hypertrophy in the hearts of METH-treated mice. Taken together, our data indicate that the cardiac pathologies associated with METH abuse following a relevant "binge and crash" METH treatment pattern in mice mimic those of its human abusers. Hence, our reported model will facilitate future studies to investigate the molecular and genetic targets underlying the development of METH-associated cardiomyopathy.

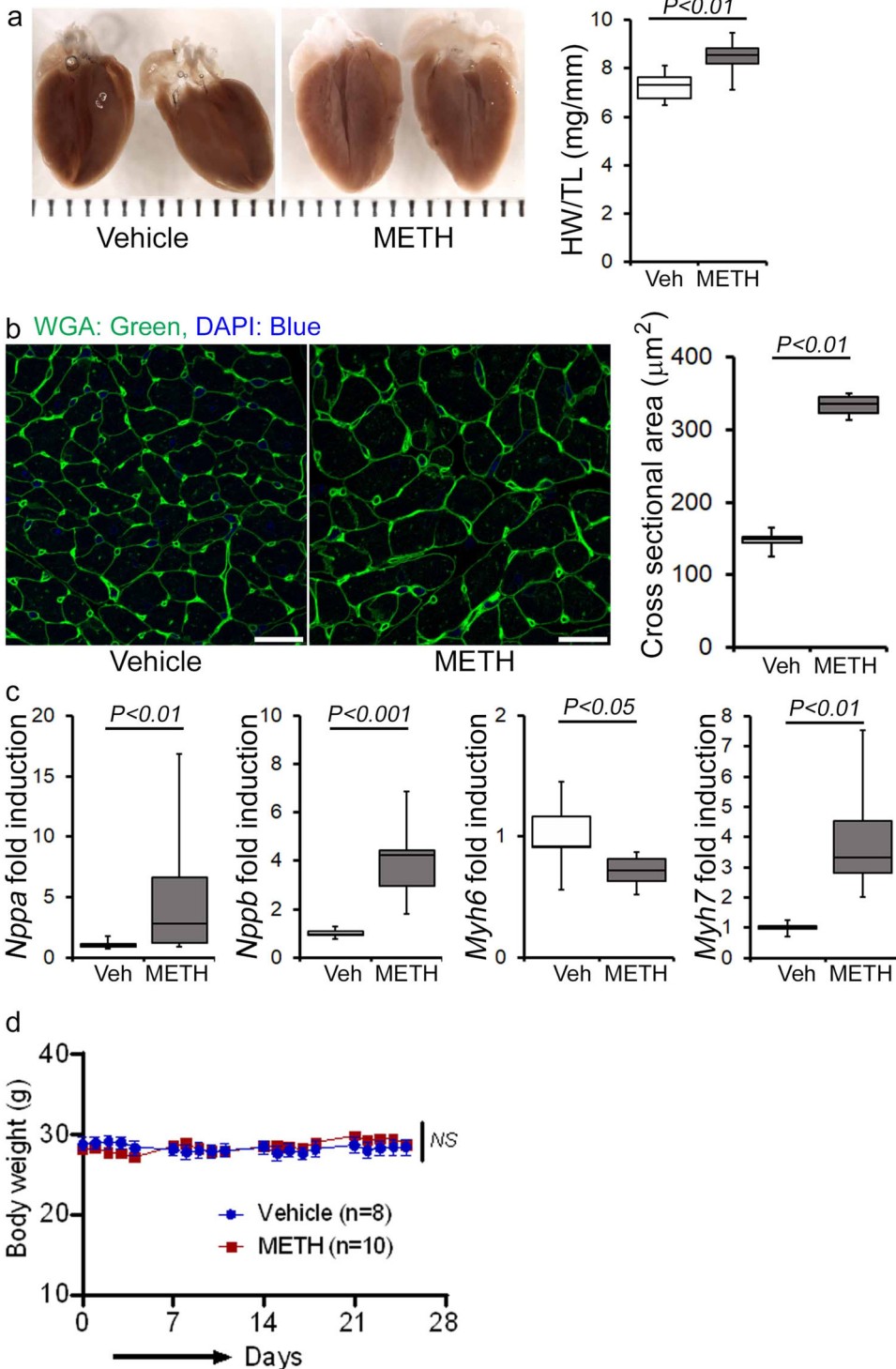

**Fig. 4 METH "binge and crash" administration induces cardiac hypertrophy in mice. a** Left panel, stereomicroscope images showing the gross morphometry of whole heart cross-sections from vehicle- and METH-treated mice. Right panel: the box diagram represents increased gravimetric heart weight (mg)-to-tibia length (cm) ratio in the METH (n = 7) group compared with the vehicle (n = 7) group. **b** Left panel: representative fluorescent images of wheat germ agglutinin stained (WGA, green) LV myocardium. Nuclei were stained with DAPI (blue). Right panel: quantification of cardiomyocyte (CM) cross-sectional areas ($\mu m^2$) showed increased CM areas in METH-treated mice (n = 1620 cardiomyocytes in six individual mice LV heart sections) compared with vehicle-treated mice (n = 1547 cardiomyocytes in five different mice LV heart sections) following 4 weeks of treatment. Scale bar: 50 $\mu$m. **c** mRNA expression of natriuretic peptide A (*Nppa*), natriuretic peptide B (*Nppb*), α-myosin heavy chain (*Myh6*), and β-myosin heavy chain (*Myh7*) expressed as the fold change in METH-treated hearts compared with vehicle-treated hearts. **d** Body weight (g) changes in vehicle (n = 8) and METH (n = 10) treated mice throughout the 4 weeks treatment period. Body weight data are expressed as mean ± SEM. Boxes depict interquartile ranges, lines represent medians, and whiskers represent ranges. *P* values were determined by two-tailed unpaired Student's *t* test for **a–c** data. Two-way mixed effects ANOVA for repeated measures followed by Sidak's multiple comparisons test was conducted to determine *P* value for **d** data. A *P* < 0.05 between groups considered statistically significant. *Veh* vehicle, *METH* methamphetamine.

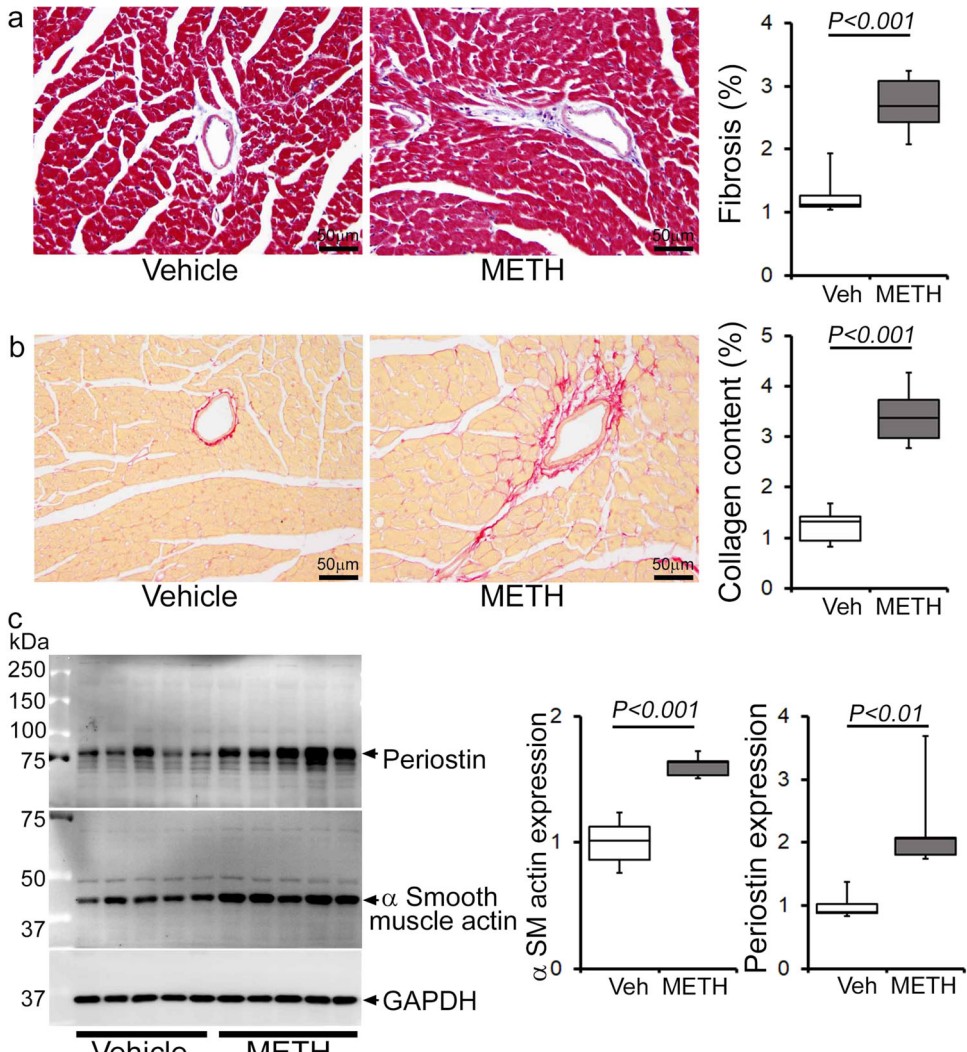

**Fig. 5 METH treatment induces cardiac fibrotic remodeling in mice.** Histopathological staining and immunoblot analysis showed increased cardiac fibrosis and enhanced expression of protein markers of fibrosis in the hearts of METH-treated mice compared with vehicle-treated mice. **a** Left panel: representative micrographs of Masson's Trichrome stained LV sections in vehicle- and METH-treated mice. Right panel: quantification of the percent myocardial fibrosis area in the vehicle-treated ($n = 5$), and METH-treated ($n = 6$) mice. Scale bar: 50 μm. **b** Left panel: representative micrographs of Picro Sirius Red-stained LV heart sections in vehicle- and METH-treated mice. Right panel: quantification of the percent collagen area in the vehicle-treated ($n = 5$) and METH-treated ($n = 6$) mice. Scale bar: 50 μm. **c** Left panel: representative immunoblot images of periostin and α-smooth muscle actin; right panel: densitometric quantification of protein bands in the hearts of vehicle- and METH-treated mice. GAPDH was used as a loading control. Boxes depict interquartile ranges, lines represent medians, and whiskers represent ranges. *P* values were determined using a two-tailed unpaired Student's *t* test. *P* < 0.05 between groups was considered statistically significant. *Veh* vehicle, *METH* methamphetamine.

Sigmar1 has been implicated as a therapeutic agent in the METH-induced addictive process and toxicity[29,53], and has been extensively studied in this regard as METH binds to Sigmar1 at physiologically relevant concentrations (Ki 2.16 ± 0.25 μM)[28]. Sigmar1 agonists have been shown to attenuate METH-induced behavioral responses, hyperthermia, and neurotoxicity[28,29,53–58]. Pretreatment with the Sigmar1 agonist PRE-084 decreases METH-induced psychomotor responses, drug-seeking behavior, and enhancement of the brain reward function[29,53]. The consequences of METH binding to Sigmar1 are currently unknown; however, studies suggest that METH may exhibit antagonist activity[23] and/or act as an inverse agonist for Sigmar1[59]. Despite extensive studies that have demonstrated Sigmar1's protective role in METH-induced cellular dysfunction[28,29,53–58], it remains unknown whether and how Sigmar1 contributes to cellular protection. Our research is the first to show that the levels of Sigmar1 expression are significantly decreased in

human METH users and in the heart tissue of "binge and crash" METH-treated mice. Moreover, the altered mitochondrial dynamics, abnormal mitochondrial ultrastructure, and impaired mitochondrial bioenergetics induced by METH treatment in mouse hearts are similar to those observed in Sigmar1[−/−] mice hearts, suggesting that Sigmar1 has a potential molecular function in the heart[32].

Similar to Sigmar1, CREB signaling has been extensively studied with respect to its relationship with METH use, and there are contradictory findings concerning CREB activation by METH exposure. We found that the expression of the phosphorylated CREB at Serine133 was significantly reduced in hearts from human METH toxicity autopsies compared with non-METH user heart tissue. Similarly, LV heart sections from our preclinical "binge and crash" METH-treated mice, as well as cultured cardiomyocytes, also showed decreased CREB phosphorylation. Our genetic studies confirm that Sigmar1 acts as an upstream regulator of CREB

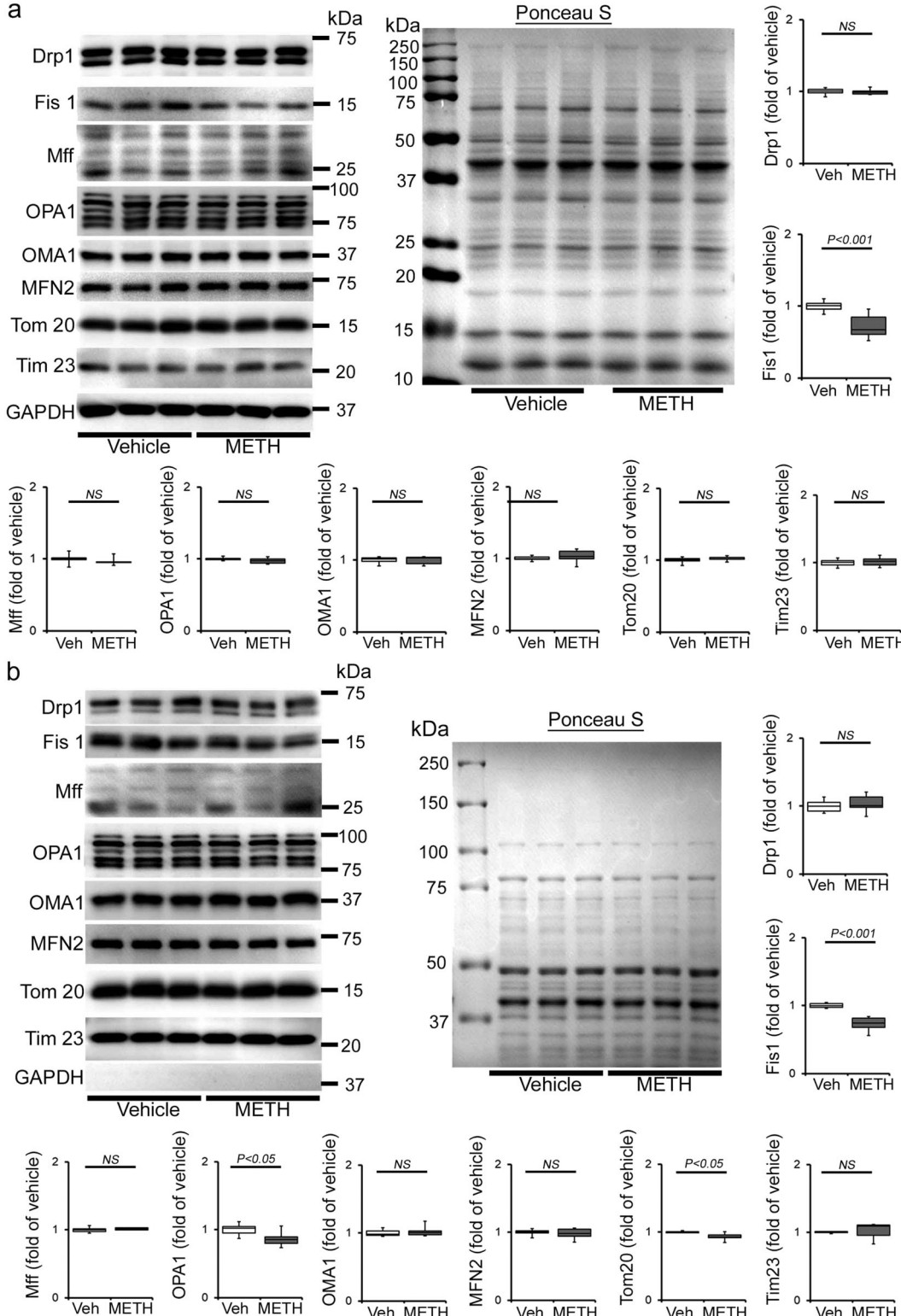

**Fig. 6 METH alters the expression of mitochondrial dynamics regulatory protein Fis1 in the heart. a**, **b** Representative immunoblot images and densitometric quantification of the expression of mitochondrial fission, i.e., Drp1, Fis1, Mff, and fusion, i.e., OPA1, OMA1, Mfn2, regulatory proteins in the whole-cell **a** and isolated mitochondrial **b** fractions from vehicle- and METH-treated mouse hearts. GAPDH was used on the same membrane of whole-cell and mitochondrial fractions to assess the purity of isolation of mitochondria. Mitochondrial outer membrane integral protein, Tom20, and mitochondrial inner membrane integral protein, Tim23, were used to confirm mitochondrial extracts. Ponceau S staining of the transfer membranes was used to verify approximately equal loading and transfer across the gel. Bar graphs represent mitochondrial dynamics regulatory protein expression changes in the hearts of METH-treated mice compared with those of vehicle-treated mice ($n = 3$–10 mice per group). Boxes depict interquartile ranges, lines represent medians, and whiskers represent ranges. $P$ values were determined using a two-tailed unpaired Student's $t$ test. $P < 0.05$ between groups was considered statistically significant. *Veh* vehicle, *METH* methamphetamine, *NS* not significant.

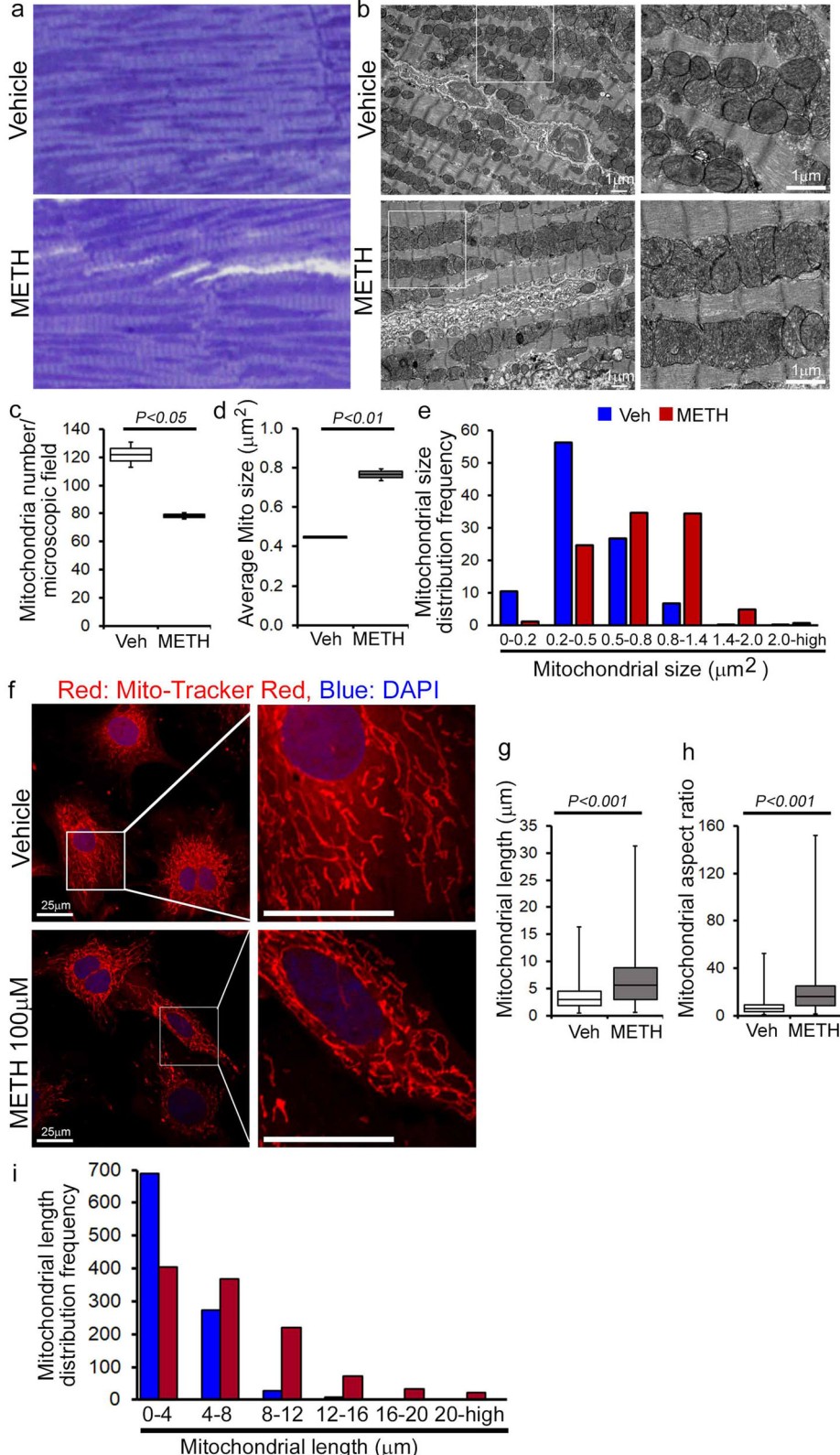

since Sigmar1 overexpression increases CREB and pCREBSer133 expression. Conversely, Sigmar1 genetic ablation attenuates CREB expression and pCREBSer133 expression. Our findings are agreement with studies that demonstrated that chronic METH treatment in rats (4 mg/kg, IP, 2 weeks) resulted in decreased CREB/pCREB immunostaining in the neuronal nuclei of striatum[60] and decreased pCREB expression in METH-treated cultured primary cortical neurons[61]. Similarly, both acute and repeated METH exposure (5 mg/kg, IP) suppresses the striatal expression of CREB mRNA[62]. Self-administration of METH (0.1 mg/kg/infusion) in rats also caused significant decreases in CREB protein levels (~40%) at 1 month in the dorsal striata[63]. pCREBSer133 levels were significantly decreased in the nucleus accumbens and ventral pallidum at 14 days withdrawal following 5 once-daily injections of 2.5 mg/kg

**Fig. 7 METH treatment alters mitochondrial morphology in mouse hearts and isolated primary cardiomyocytes.** Representative **a** toluidine blue-stained and **b** transmission electron micrographs of vehicle- and METH-treated mouse LV heart sections show the preponderance of large, abnormally shaped mitochondria in the hearts of METH-treated mice. Scale bar: 1 μm. Bar graphs represent **c** the number of mitochondria per microscopic field, **d** average mitochondrial size ($\mu m^2$), and **e** mitochondrial size ($\mu m^2$) distribution frequency. Mitochondrial areas ($\mu m^2$) were measured in 20 microscopic fields containing 2436 mitochondria from vehicle-treated mouse hearts ($n = 2$) and in 24 microscopic fields containing 1879 mitochondria from METH-treated mouse hearts ($n = 2$). **f** Representative fluorescent micrographs of Mitotracker Red staining used to visualize mitochondrial network morphology in cultured cardiomyocytes (NRC) following treatment with vehicle or METH (100 μM) for 24 hours. METH treatment resulted in visually distinct hyperfused mitochondria in cardiomyocytes compared with the mitochondria observed in the vehicle-treated group. Bar graphs represent **g** average mitochondrial network length (μm), **h** mitochondrial aspect ratio, and **i** mitochondrial length distribution frequency. Mitochondrial length (μm) and width (μm) were measured (>950 mitochondria for each treatment group) from three independent experiments in the cardiomyocytes of the vehicle- and METH-treated mice using NIH ImageJ (v1.6.0) software. Boxes depict interquartile ranges, lines represent medians, and whiskers represent ranges. P values were determined using a two-tailed unpaired Student's t test. P < 0.05 between groups was considered statistically significant. *Veh* vehicle, *METH* methamphetamine.

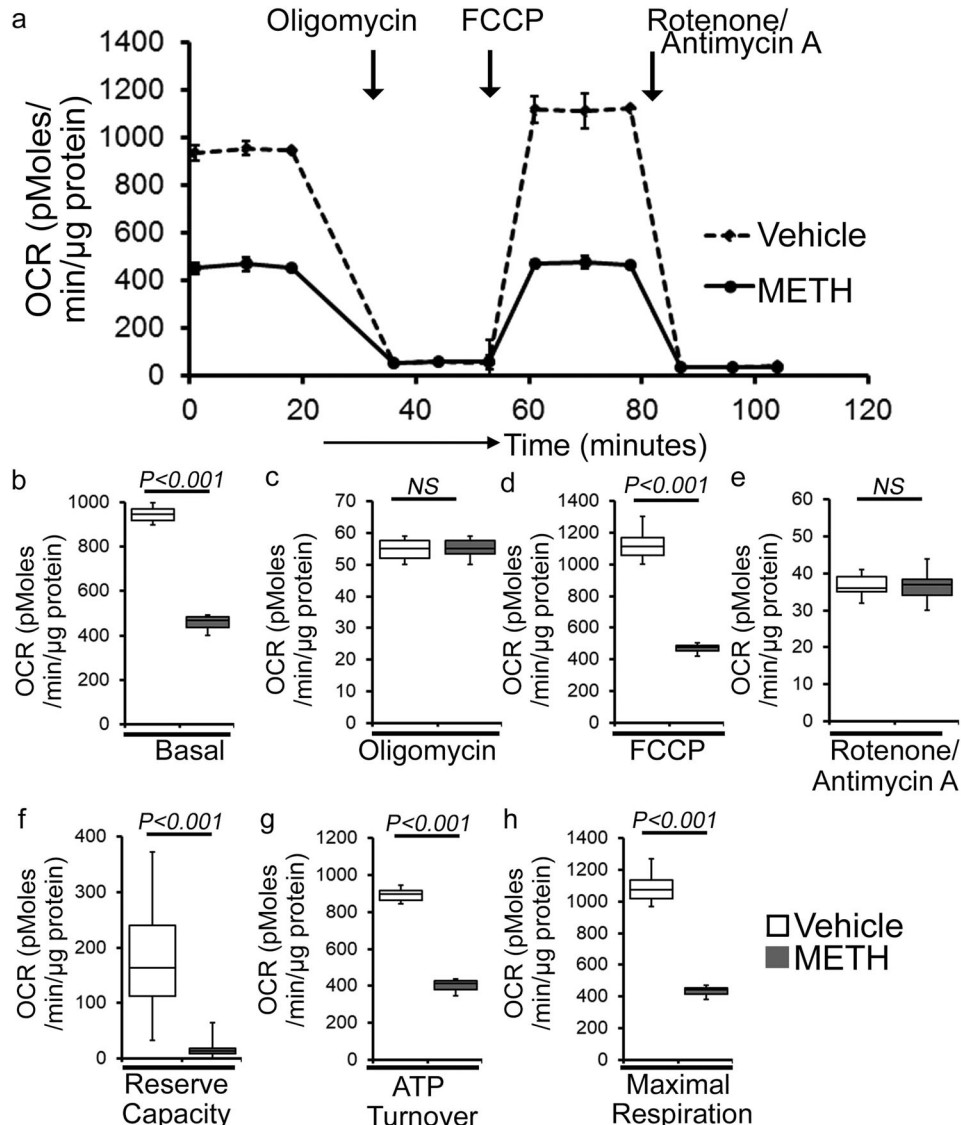

**Fig. 8 METH treatment induces impaired mitochondrial bioenergetics in the heart. a** Summary traces of mitochondrial oxygen consumption rate (OCR) profiles as measured in isolated mitochondria from the hearts of vehicle- and METH-treated mice. The black arrow indicates the sequential addition of oligomycin (1 μmol/L), FCCP (carbonyl cyanide 4-(trifluoromethoxy) phenylhydrazone) (4 μmol/L), and rotenone (0.5 μmol/L) plus antimycin A (0.5 μmol/L). OCR profiles are expressed as pmol $O_2$/[min*μg protein], and each point represents average OCR values from five individual mouse cardiac mitochondria per group. **b**–**e** Bar graphs represent OCR under **b** baseline and with the addition of **c** oligomycin, **d** FCCP, and **e** rotenone plus antimycin A. **f**–**h** Key parameters of mitochondrial bioenergetics, including **f** reserve capacity, **g** ATP turnover, and **h** maximal respiration were significantly decreased in METH-treated mice compared to vehicle-treated mice. Boxes depict interquartile ranges, lines represent medians, and whiskers represent ranges. P values were determined using a two-tailed unpaired Student's t test. P < 0.05 between groups was considered statistically significant. *Veh* vehicle, *METH* methamphetamine.

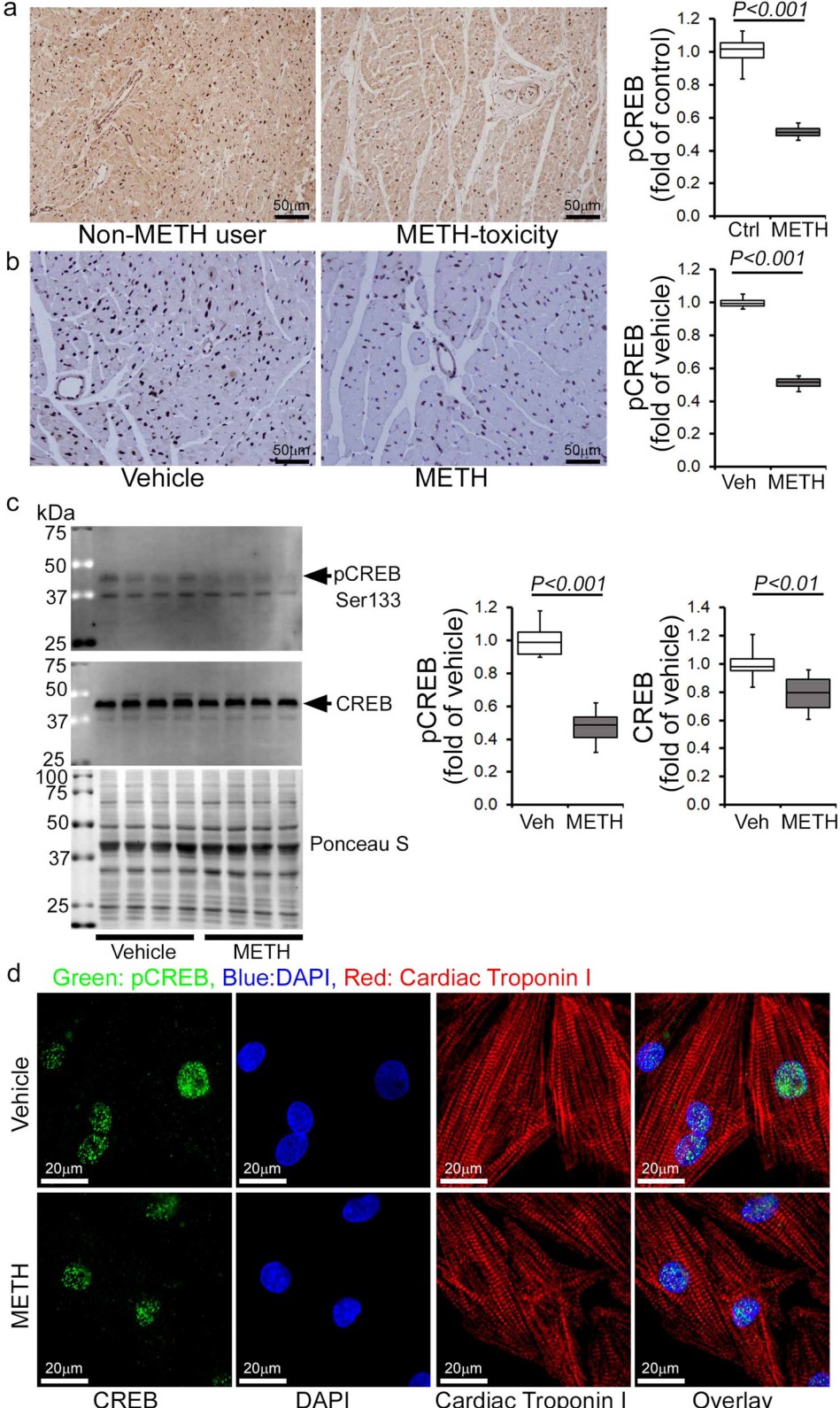

METH[64]. In contrast, increased pCREB phosphorylation has also been reported in the self-administered METH rat striatum[65]. Despite the data reported in all of these studies, CREB nuclear translocation may be an early event activated by METH as observed immediately after METH treatment (30 mg/kg, 30 min) in mice and METH exposure (150 μM) in primary rat astrocytes[66]. Moreover, Sigmar1 regulates the CREB translocated into nucleus, as

pretreatment with the Sigmar1 antagonist in astrocytes blocked CREB nuclear translocation[66]. The Sigmar1 gene has a CREB-binding site in its promoter region; METH-mediated activation of astrocytes involved CREB translocated into the nucleus and interaction with the promoter of Sigmar1, resulting in increased expression of Sigmar1[66]. Similarly, Sigmar1 upregulation was observed in cultured astrocytes from adult rats after infection with

**Fig. 9 METH treatment attenuates CREB expression and phosphorylation in human and mouse hearts.** Immunohistochemistry and immunoblot based analyses showed that METH induces a marked reduction of phosphorylation of CREB at the Serine133 residue in both human and mouse hearts. **a** Left panel: representative micrographs of immunohistochemical detection of phosphorylated CREB at Ser133 (brown color, positive DAB staining) in human non-METH user ($n = 4$) and METH toxicity ($n = 4$) LV sections. Hematoxylin (blue) was used to counterstain nuclei. Right panel: the bar graph represents the fold change of DAB-positive pCREBSer133 staining intensity in the nucleus of METH toxicity human hearts compared with non-METH user human hearts. Scale bar: 50 μm. **b** Left panel: representative micrographs of immunohistochemical detection of phosphorylated CREB at Ser133 (brown, DAB staining) in the vehicle-treated ($n = 4$) and METH-treated ($n = 4$) mouse LV heart sections. Hematoxylin (blue) was used to counterstain the nuclei. Right panel: the bar graph represents the fold change of DAB-positive pCREBSer133 staining intensity in the nucleus of the hearts of METH-treated mice compared to vehicle-treated mice. Scale bar: 50 μm. **c** Immunoblot analysis of total CREB and phosphorylated CREB at Ser133 (pCREBSer133) in vehicle- and METH-treated mouse heart-whole-cell lysates. Ponceau S staining of the transfer membranes was used to confirm approximately equal loading and transfer across the gel. Bar graphs represent densitometric quantification of total CREB and pCREBSer133 expression in the hearts of vehicle- and METH-treated mice ($n > 5$ mice per group). Boxes depict interquartile ranges, lines represent medians, and whiskers represent ranges. *P* values were determined using a two-tailed unpaired Student's *t* test. $P < 0.05$ between groups was considered statistically significant. **d** Immunofluorescent micrographs of pCREBSer133 staining (green) in vehicle- and METH-treated (100 μM) neonatal rat cardiomyocytes (NRC) show marked reduction of pCREBSer133 in cardiac Troponin I (red)-positive cardiomyocytes. DAPI (blue) was used to counterstain the nuclei. Images are representative of three independent experiments. Scale bar: 20 μm. *Veh* vehicle, *METH* methamphetamine.

the VP16-CREB viral vector, and in transgenic mice with targeted activation of CREB in astrocytes[67]. All of these studies suggest a CREB-Sigmar1 feedback loop regulating their own expression modulated by METH exposure. Future studies are required to determine the precise molecular mechanisms of Sigmar1 down-regulation in cardiomyocytes mediated by METH exposure.

The physiological importance of CREB signaling in maintaining normal cardiac function is demonstrated by studies showing that mice bearing cardiomyocyte-specific dominant-negative CREB deficient in phosphorylation develop severe and progressive dilated cardiomyopathy with mitochondrial dysfunction[68,69]. Moreover, pathologic stress, including hypoxia and high-glucose conditions, has been shown to regulate CREB stability and expression[70,71]. Our findings showed a positive correlation between mitochondrial dysfunction and decreased CREB activity as a result of METH treatment. METH-treated mouse heart tissue and cardiomyocytes showed altered mitochondrial dynamics associated with decreased Fis1 expression and defects in mitochondrial respiration. Fis1-dependent mitochondrial fission has been reported in the heart; it does not alter the level of classical cytosolic Drp1 translocation to the mitochondria[42,72] and is reported to be independent of putative cytosolic mitochondrial fission, promoting GTPases such as Drp1 and Dyn2[72]. Our findings indicate that METH attenuates Fis1 protein expression in both the whole-cell fraction and the mitochondrial fraction. METH may alter Fis1 expression via inhibition at the transcriptional level, as supported by a recent elegant study showing that CREB binding to the Fis1 gene promoter region promotes Fis1 expression in cardiomyocytes[42]. Sigmar1-dependent regulation of Fis1 expression was also observed in Sigmar1 over-expression and siRNA knockdown cardiomyocytes with a positive correlation with CREB activity. Our study suggests that Sigmar1-dependent modulation of the CREB-Fis1 signaling axis may play a role in regulating mitochondrial morphology and function.

In conclusion, METH induces adverse cardiac fibrotic remodeling in human METH users and preclinical "binge and crash" METH-treated mouse heart. We demonstrated a novel correlation among Sigmar1 level, Sigmar1-dependent regulation of the CREB-Fis1 signaling axis, and mitochondrial dysfunction contributing to the development of METH-associated cardiomyopathy. Clinical translation of these findings would require the completion of critical studies using genetic mice models of Sigmar1 to provide further evidence for the therapeutic efficacy of Sigmar1. Our ongoing studies are focused on defining the molecular role of Sigmar1 in regulating cardiac CREB expression and activation using genetic, molecular, and biochemical experimentation involving cardiac-specific Sigmar1 transgenic and knockout mice.

## Methods

**Human heart samples**. We collected a total of 34 human heart sections from autopsies of patients who were positive for METH following toxicological studies. We also collected heart sections from autopsies from seven non-METH users who were negative for METH and other drugs of abuse following toxicological studies. All heart specimens were obtained in collaboration with the Department of Pathology and Translational Pathobiology at LSU Health Sciences Center-Shreveport. The LV free walls were dehydrated, paraffin-embedded, and serial 5 μm thin sections were cut for subsequent histological studies. All experiments using human tissue was deemed non-human research by the local IRB owing to the exclusive use of postmortem samples.

**Animals**. We used 8- to 10-week-old male C57BL/6 mice (Jackson Laboratory, Bar Harbor, ME) to study the effects of "binge and crash" METH administration on cardiac pathology. Mice were housed in cages supplied with ad libitum water and food with a 12-hour light–dark cycle. We acquired timed pregnant Sprague-Dawley rats from Charles River Laboratories (Portage, MI) to isolate primary NRCs from newborn rat pups. All animals were handled and cared for according to the Guide for the Care and Use of Laboratory Animals: Eighth Edition (National Institutes of Health, Bethesda, MD). All animal experiments were approved by the Institutional Animal Care and Use Committee of LSU Health Sciences Center-Shreveport.

**"Binge and crash" METH-dosing protocol**. "Binge and crash" use of METH is a prevalent form of chronic METH abuse in human METH users involving several drug administrations in a dose-escalation manner on a single day for several days ("binge" period) followed by a period of abstinence. To mimic the "binge and crash" pattern of METH abuse in humans, we have adopted a published "binge and crash" METH treatment model in mice[34]. To induce "binge and crash" METH treatment, we randomly allocated 8- to 10-week-old male C57BL/6 mice to receive 0–6 mg/kg METH (methamphetamine HCl, Sigma-Aldrich, St. Louis, MO) via subcutaneous (s.c.) injection 5 days a week for 4 weeks as detailed in Supplementary Table 2. METH was dissolved in sterile saline solution (Sigma-Aldrich; 0.9% w/v NaCl). Vehicle-treated mice received the same volume of saline at all-time points for 4 weeks. All data were acquired following 4 weeks of experimental vehicle and METH administration, as outlined below.

**Histological studies**. Human and mice paraffin-embedded heart sections were cut in serial 5 μm thin sections, deparaffinized, hydrated, and stained with PSR and Masson's Trichrome to assess the extent of collagen deposition and fibrotic area, respectively[32,36–38]. All stained heart sections were numbered in an alphanumeric fashion and imaged on an Olympus BX40 microscope using bright field mode with a ×10 (human heart sections) and ×20 (mice heart sections) objective lens in an investigator-blinded manner. Measurements were made of the percent collagen deposition (stained red with PSR) and fibrosis area (μm$^2$) (stained blue with Masson's Trichrome), respectively, as a total of the stained myocardium area (μm$^2$) within each microscopic field using the National Institutes of Health (NIH) ImageJ (v1.6.0) software (Bethesda, MD)[32,37]. All quantifications were analyzed in a blinded manner using randomly selected 10–15 high-magnification microscopic fields from LV for each human and mouse heart section.

**Echocardiography**. Cardiac functional and structural parameters were measured in isoflurane-anesthetized mice and quantified using M-mode echocardiography images. The images were captured using Vevo LAB 3.1.1 software (VisualSonics) acquired by a VisualSonics Vevo 3100 high-resolution ultra-sound imaging system (Toronto, ON, Canada) with a 40-MHz transducer[32,35–38]. Two-dimensional parasternal short axis M-mode images were analyzed using Vevo LAB software to

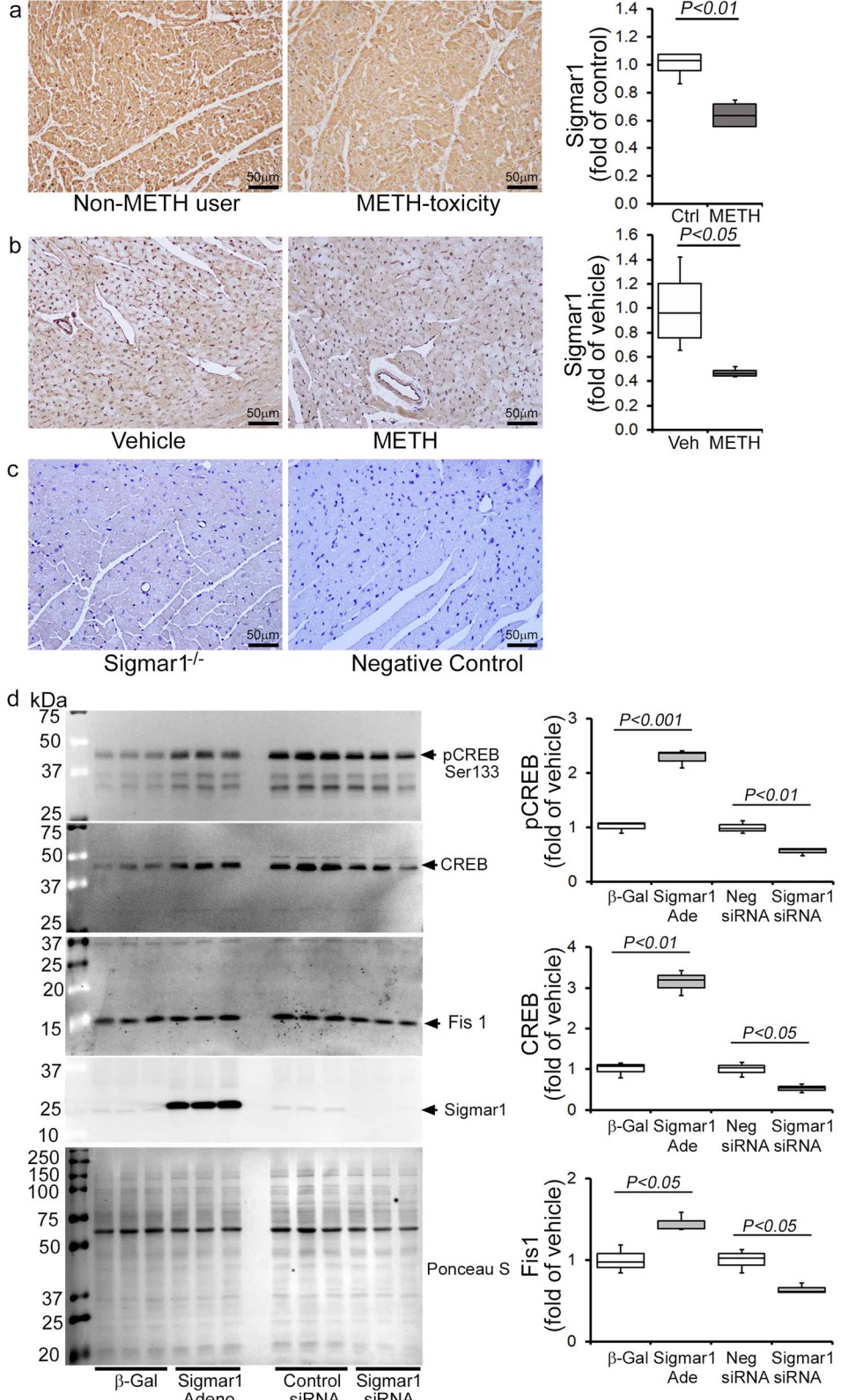

quantify LV structural indices including LV internal diameter at systole and diastole (LVID;s and LVID;d, respectively), LV volume at systole and diastole (LV Vol;s and LV Vol;d, respectively), interventricular septum thickness at systole and diastole (IVS;s and IVS;d, respectively), LV posterior wall thickness at systole and diastole (LVPW;s and LVPW;d, respectively), and LV mass. Percent fractional shortening (%FS = (LVID;d-LVID;s)/LVID;d × 100) and percent ejection fraction (%EF = (LV Vol;d-LV Vol;s)/LV Vol;d × 100) were calculated as systolic cardiac functional indices in vehicle- and METH-treated mice.

**Heart gravimetric studies**. Whole hearts were excised from the mice following sacrifice, and any excess blood was soaked up using clean paper towels. The wet weights were measured using an OHAUS® electronic balance machine (NJ, USA). Heart gravimetric indices were measured by comparing the heart weight (mg)-to-tibia length (cm) ratios in vehicle- and METH-treated mice.

**WGA staining**. To assess cardiomyocyte hypertrophy, serial 5 μm paraffin heart sections were deparaffinized, hydrated, antigen retrieved, and stained with Alexa

**Fig. 10 METH treatment causes Sigmar1 expression-dependent regulation of CREB-Fis1 expression in cardiomyocytes. a** Left panel: representative micrographs of immunohistochemical detection of Sigmar1 (brown) in human non-METH user ($n = 4$), and METH toxicity ($n = 4$) LV heart sections. Hematoxylin (blue) was used to counterstain the nuclei. Right panel: fold change of DAB-positive Sigmar1 staining intensity in METH toxicity hearts compared to those of non-METH users. **b** Left panel: representative micrographs of immunohistochemical detection of Sigmar1 (brown) in the vehicle ($n = 4$), and METH-treated ($n = 4$) mouse LV sections. Right panel: fold change of DAB-positive Sigmar1 staining intensity in hearts treated with METH hearts compared with those treated with vehicle. **c** Left panel: global Sigmar1 knockout (Sigmar1$^{-/-}$) mouse hearts were used to assess the specificity of the Sigmar1 primary antibody. Right panel: IgG stained slides without primary antibody incubation were used as a negative control. Representative micrographs are from three independent experiments. **a–c** Scale bar: 50 μm. **d** Sigmar1 was overexpressed and knocked down in NRC through adenovirus infection (10 MOI) and siRNA (50 nM) transfection, respectively. Right panel: western blot analyses showed that Sigmar1 overexpressed NRC exhibited increased CREB, pCREBSer133, and Fis1 expression, whereas knockdown of Sigmar1 yielded opposite results. Ponceau S staining of the transfer membranes was used to confirm equal loading and transfer across the gel. Left panel: bar graphs represent densitometric quantification of CREB, pCREBSer133, and Fis1 protein bands in cardiomyocytes. β-Gal adenovirus infection was used as the control for Sigmar1 adenovirus infection, whereas a non-specific siRNA was used as the control for Sigmar1 siRNA transfection. Data are representative of triplicates from two independent experiments. Boxes depict interquartile ranges, lines represent medians, and whiskers represent ranges. *P* values were determined using a one-way ANOVA, followed by Tukey's multiple comparisons test. *P* < 0.05 between groups was considered statistically significant. *β-Gal* β-galactosidase; *Adeno* adenovirus, *siRNA* small interfering RNA, *Veh*, vehicle, *METH* methamphetamine, *NS* not significant.

Fluor 488 WGA (5 μg/mL, Invitrogen) for 1 hour at room temperature (RT)[32]. Nuclei were counterstained with 4',6-diamidino-2-phenylindole (DAPI, Invitrogen). Stained heart sections were visualized and imaged on a Nikon A1R high-resolution confocal microscope (Nikon Instruments Inc., Melville, NY) coupled with Nikon NIS elements software (v4.13.04) with a ×60 oil objective lens (NA = 1.4). The cross-sectional areas (μm$^2$) of the cardiomyocytes from vehicle- and METH-treated mice were measured in an analyst-blinded manner on alphanumerically labeled images to calculate the average area using NIH ImageJ software.

**RNA isolation and quantitative real-time polymerase chain reaction analysis**. Total RNA was isolated from flash-frozen heart samples using RNAzol RT (Molecular Research Center, Cincinnati, OH) according to the manufacturer's protocol[32,36–38]. Complementary DNA (cDNA) was synthesized using iScript™ Advanced cDNA synthesis kit for RT-qPCR (Bio-Rad, Hercules, CA) using 5 μg RNA as templates. We performed quantitative real-time polymerase chain reaction (PCR) on a CFX96 Touch™ Real-time PCR detection system (Bio-Rad) using TaqMan® probes (Applied Biosystems, Foster City, CA) for fetal cardiac genes including *Nppa*, *Nppb*, *Myh6*, and *Myh7* as described earlier. β-actin (Applied Biosystems) was used as a housekeeping gene[36]. All samples were analyzed in triplicate to calculate the fold change in mRNA expression using the comparative $C_t$ method [$2^{\wedge}(-\Delta\Delta C_t)$]. Values were averaged and expressed as the fold change in mRNA expression in METH-treated mice compared to those in vehicle-treated mice.

**Heart mitochondria isolation**. Mitochondria were isolated from the freshly excised hearts of anesthetized mice. Harvested heart ventricles were homogenized in mannitol-sucrose-ethylene glycol tetraacetic acid (EGTA) buffer containing 225 mM mannitol, 75 mM sucrose, 5 mM HEPES, and 1 mM EGTA (pH 7.4) using a glass/Teflon Potter Elvehjem homogenizer[32,35,38]. Mitochondria were isolated from the homogenized heart lysates using differential centrifugation. Fresh mitochondria were subjected to a mitochondrial bioenergetics stress test, as outlined below. Mitochondria for western blot analyses were lysed in Cell Lytic M buffer (Sigma-Aldrich) supplemented with protease and phosphatase inhibitors (Roche).

**NRC isolation and culture**. We isolated primary neonatal rat ventricular cardiomyocytes from 1- to 2-day-old Sprague-Dawley rat pups[35,38,73]. The excised rat pup heart ventricles underwent sequential enzymatic digestion with Collagenase II (Worthington) and Trypsin (Gibco). Cardiac fibroblasts were cleared through a pre-plating step. Next, the isolated cardiomyocytes were plated at $1.5 × 10^6$ cell density per 10 cm$^2$ plates in αMEM media containing 10% fetal bovine serum (FBS, Gibco) supplemented with 1% antibiotic–antimycotic mixture (Gibco). After 24 hours, the medium was changed to Dulbecco's Modified Eagle's medium (DMEM, Gibco) containing 2% FBS and 1% antibiotic–antimycotic. The cardiomyocytes were subjected to treatment with the vehicle or METH at the concentration and time points indicated.

**Mitochondrial functional assessment**. We conducted a mitochondrial bioenergetics stress test to measure the mitochondrial OCR using a Seahorse XFe24 Extracellular Flux Analyzer (Agilent, Santa Clara, CA). Cardiac mitochondria isolated as described above were seeded in aliquots of 50 μg protein per well in XFe24 Seahorse plates in respiration media (pH 7.4) containing 220 mM mannitol, 70 mM sucrose, 10 mM KH$_2$PO$_4$, 5 mM MgCl$_2$, 2 mM HEPES, 1 mM EGTA, and 0.2% w/v fatty acid-free bovine serum albumin supplemented with 7 mM pyruvate and 1 mM malate. The mitochondrial OCR was measured at baseline and following sequential addition of 1 μM oligomycin (ATP synthase inhibitor), 4 μM carbonyl cyanide 4-FCCP) (a mitochondrial uncoupler, and 0.5 μM rotenone (Complex I

inhibitor) plus 0.5 μM antimycin A (Complex III inhibitor)[32,35,38]. The OCR values were normalized to total protein content in the respective wells and expressed as pmol/min/μg protein.

To measure OCR in intact neonatal rat ventricular cardiomyocytes (NRC), NRC was seeded at a density of $8 × 10^4$ cells per well into 0.1% w/v gelatin-coated Seahorse XFe24 plates with DMEM maintenance media containing 2% FBS with 1% antibiotic–antimycotic mixture (Gibco). After 24 hours of stabilization, the cardiomyocytes were treated with vehicle, METH or NE dissolved in culture media for another 24 hours. The NRC were then incubated with DMEM (containing no glucose and no pyruvate; Gibco) containing 10 mM glucose and 2 mM sodium pyruvate in a CO$_2$ free incubator at 37 °C for 1 hour before loading the plate in the XFe24 analyzer. The OCR were measured over 90 minutes while oligomycin (1 μM), FCCP (4 μM), and rotenone (0.5 μM) plus antimycin A (0.5 μM) were added sequentially to each well at the time points given. Upon completion of the test protocol, total protein concentrations were measured in individual wells and OCR values were expressed as pmol/min/μg protein[35,38].

**Western blot analyses**. Whole-cell heart tissue lysates were prepared by homogenizing flash-frozen hearts in ice-cold Tris-HCl buffer (pH 7.5) containing 0.5% Triton-X 100, 4 mM EGTA, 10 mM ethylenediaminetetraacetic acid (EDTA), 150 mM sodium chloride, 1 mM sodium orthovanadate, 30 mM sodium pyrophosphate, 50 mM sodium fluoride, and a complete mixture of protease inhibitor cocktail[32,35,38]. The lysed heart homogenates were cleared by centrifugation at $12,000 × g$ for 15 minutes to pellet the insoluble materials. The supernatant was used for subsequent western blot experiments.

NRC and HEK293T cells were lysed with Cell Lytic M lysis buffer (Sigma-Aldrich) containing a protease inhibitor cocktail (Roche). The lysed cells were sonicated and centrifuged at $14,000 × g$ for 15 minutes to sediment insoluble cell debris. The protein content of the soluble heart tissue fraction and NRC lysates was quantified using modified Bradford reagent (Bio-Rad) reactive to a bovine serum albumin (BSA) standard curve (Bio-Rad). Equal amounts of protein (10–20 μg) in 1× Laemmli's buffer were separated on sodium dodecyl sulfate polyacrylamide gel electrophoresis using 7.5–12% gels and transferred to polyvinylidene difluoride (PVDF) membranes (Bio-Rad). Membranes were blocked with 5% non-fat dried milk for 1 hour and incubated with primary antibodies overnight at 4 °C. The following primary antibodies were used for immunoblotting: Drp1 (1:1000, 14647, Cell Signaling), Fis1 (1:500, sc-98900, Santa Cruz), Mff (1:500, sc-398617, Santa Cruz), OPA1 (1:1000, 80471, Cell Signaling), Mfn2 (1:1000, 9482, Cell Signaling), Tom20 (1:1000, sc-11415, Santa Cruz), Tim23 (1:500, sc-514463, Santa Cruz), CREB (1:500, 9197, Cell Signaling), pCREBSer133 (1:500, 9198, Cell Signaling), Sigmar1 (1:1000, 61994, Cell Signaling), and GAPDH (1:1000, MAB374, EMD Millipore). Membranes were washed with 1× TBS-T buffer, incubated with alkaline-phosphatase-conjugated secondary antibodies (Jackson ImmunoResearch Laboratories, Inc.), and imaged on a ChemiDoc™ Touch Imaging System (Bio-Rad) following being developed using ECF substrate (Amersham). Ponceau S protein staining was used on the transfer PVDF membrane to confirm equal protein loading. The intensity of the protein bands was quantified using densitometric analysis with NIH ImageJ software.

**Immunohistochemistry**. Human and mouse heart sections (5 μm) were deparaffinized, hydrated, and heat-induced antigen retrieved with antigen unmasking solution (H-3301, Vector Laboratories, Burlingame, CA). Endogenous peroxidases were blocked by incubation with 0.3% v/v hydrogen peroxide. Sections were blocked with 5% serum (Vector Laboratories) for 1 hour at RT. Following blocking, sections were incubated with primary antibodies diluted in blocking serum overnight in a humid staining box at 4 °C. The following primary antibodies were used in this study: Sigmar1 (1:100, 61994, Cell Signaling) and pCREBSer133 (1:100,

9198, Cell Signaling). The antigen-antibody interaction was amplified and visualized using the VECTASTAIN Elite ABC Peroxidase kit (Vector Laboratory) following the manufacturer's protocol with the addition of the DAKO DAB-chromogen System (DAKO, Carpinteria, CA)[74]. Sections were then counterstained with hematoxylin, dehydrated, and cover-slipped with Cytoseal XYL mounting medium (Thermo Scientific). Stained heart sections were examined under an Olympus BX40 microscope using bright field mode with ×10 (human heart sections) and ×20 (mouse heart sections) objective lenses with Olympus cellSens imaging software (Olympus Life Science, Waltham, MA). The DAB-precipitated brown color positive staining intensity was quantified in 10 high-magnification microscopic fields per heart section using NIS Elements software (v4.13.04), averaged, normalized to control groups, and expressed as mean ± SEM.

**Immunocytochemistry**. Isolated NRCs were plated on Lab-Tek II chamber slides (Thermo Scientific, 154461) at a seeding density of $1 × 10^5$ cells per well. Next, the NRC were treated with vehicle or METH (100 μM) dissolved in NRC culture media (DMEM containing 2% FBS and 1% antibiotic–antimycotic mixture; Gibco) for 24 hours. Cells were fixed and permeabilized in 4% paraformaldehyde with 0.5% Triton-X 100 in PBS for 10 minutes. Next, the formaldehyde was quenched by incubating the cells in 0.1 M glycine solution (pH 3.5) for 30 minutes at RT. The cells were then blocked with 1% BSA, 0.1% cold water fish skin gelatin, 0.1% Tween-20, and 0.05% sodium azide in 1× PBS for 1 hour at RT. Following blocking, the cells were incubated with rabbit anti-pCREBSer133 antibody (1:100, 9198, Cell Signaling) and mouse anti-cardiac Troponin I (1:1000, MAB1691, EMD Millipore). Cells were then washed and incubated with anti-rabbit and anti-mouse secondary antibodies conjugated with Alexa 488 and Alexa 568 dyes (Invitrogen), respectively, for 1 hour at RT. Next, cells were washed with PBS, counterstained with 4′-6-diamidino-2-phenylindole (Invitrogen), and mounted with Vectashield Hardset mounting media (Vector Laboratories). Cardiomyocytes were subsequently examined using a Nikon A1R high-resolution confocal microscope (Nikon Instruments Inc., Melville, NY) and imaged with Nikon NIS elements software (v4.13.04) with a ×60 oil objective lens (NA = 1.4)[35,73]. All image acquisition was performed in an investigator-blinded manner.

**Transmission electron microscopy**. Vehicle- and METH-treated mouse LV heart sections were fixed with 2% glutaraldehyde and post-fixed in 1% OsO₄ for thin sectioning. Multiple sections were counterstained with uranium and lead salts followed by examination with a JEOL 1400 transmission electron microscope. Images were acquired with an AMT digital camera. We used two vehicle-treated and two METH-treated mouse hearts for our electron microscopy studies. Mitochondrial areas ($μm^2$) were quantified in an investigator-blinded manner in 20 microscopic fields containing 2436 mitochondria (vehicle hearts) and 24 microscopic fields containing 1879 mitochondria (METH hearts) to calculate the average mitochondrial size and determine the number of mitochondria per microscopic field using NIH ImageJ (v1.6.0) software[32].

**Mitochondrial staining**. To visualize the mitochondrial network structure in cardiomyocytes, NRCs were seeded on Lab-Tek II chamber slides (Thermo Scientific, 154461) at a density of $1 × 10^5$ cells per well. NRCs were then treated with vehicle or METH (100 μM) for 24 hours. Next, the NRCs were loaded with fixation-insensitive MitoTracker® Red CMXRos (200 nM; M7512, Invitrogen) dissolved in NRC culture medium for 30 minutes followed by immediate fixation with 4% paraformaldehyde for 10 minutes according to the manufacturer's instructions (Molecular Probes, Invitrogen). Cells were subsequently washed, counterstained with DAPI (Invitrogen), and mounted with Vectashield Hardset antifade mounting media for fluorescence (Vector Laboratories). Cardiomyocytes were subsequently visualized and imaged using a Nikon A1R high-resolution confocal microscope (Nikon Instruments Inc., Melville, NY) coupled with Nikon NIS elements software (v4.13.04) with a ×60 oil objective lens (NA = 1.4). The mitochondrial network lengths (μm) and widths (μm) from three independent experiments were measured in the vehicle-treated (998 mitochondrial networks, 42 cells) and METH-treated (100 μM; 1120 mitochondrial network, 65 cells) cardiomyocytes in 10 high-magnification microscopic field images using NIH ImageJ (v1.6.0) software as described earlier[35]. The mitochondrial aspect ratio was calculated from the ratio of length and width[40,41]. All image analyses were performed in alphanumerically labeled images in an analyst-blinded manner.

**Recombinant Sigmar1 adenovirus infection and siRNA knockdown of Sigmar1**. To overexpress Sigmar1, primary NRCs were infected with Sigmar1 carrying adenoviral vector (10 MOI) for 2 hours in DMEM media only (without serum or antibiotics), after which the media were changed to regular culture media. We prepared adenoviral constructs containing wild-type Sigmar1 by cloning into a pShuttle-CMV vector; replication-deficient recombinant adenoviruses were made using the AdEasy system (Agilent Technologies)[73]. Plates infected with adenovirus expressing β-galactosidase served as controls for all the experiments.

To knockdown Sigmar1, Sigamr1 siRNA (GGAUCACCCUGUUUCUGACU AUUGU and ACAAUAGUCAGAAACAGGGUGAUCC) (Invitrogen) was transfected into NRC using Lipofectamine 2000 (Invitrogen) in OptiMEM media (Gibco) for 16 hours[73]. Media were changed to regular culture media following

16 hours. A non-specific siRNA was used as a negative control in all silencing experiments. After 72 hours of adenovirus infection and siRNA transfection in both Sigmar1 overexpressed and knockdown experiments, NRCs were lysed using Cell Lytic M buffer (Sigma-Aldrich) containing protease and phosphatase inhibitors (Roche) for western blot analyses.

**Flow cytometry**. Flow cytometry analysis of spleen and blood was performed as described previously[75]. Spleens were homogenized in fluorescence-activated cell sorting (FACS) wash buffer (1% bovine serum albumin, 1 mM EDTA, and 0.1% sodium azide in phosphate-buffered saline). The spleens were strained with a 40 μm cell strainer (Falcon, 352340) followed by centrifugation at $300 × g$ for 5 minutes. The supernatant was decanted, and cells were dislodged in 3 mL ACK lysis buffer (0.15 M NH4Cl, 10 mM KHCO3, 0.1 mM NA2EDTA, adjusted to pH 7.2 and filter sterilized using a 0.22 μm filter). Cells were incubated on ice for 5 minutes. Cells were washed in FACS wash buffer with centrifugation. The pellet was re-suspended in 10 mL RPMI 1640 (HyClone, SH30027.01), strained (Falcon, 352340), and counted. Cells were adjusted to $5 × 10^6$ cells/mL in RPMI. Aliquots of 50 μL of blood were lysed in 3 mL ACK lysis buffer. Cells (100 μL) were mixed with 100 μL of blocking buffer (CD16/32 diluted in FACS wash buffer) and incubated at 4 °C for 20 minutes. Cells were centrifuged at 1500 rounds per minute for 5 minutes and the supernatant discarded. Primary labeled antibodies (50 μL per well) were added, light protected, and incubated at 4 °C for 30 minutes. Cells were washed three times in FACS wash buffer with centrifugation. Cells were re-suspended in cold FACS/fix solution (FACS wash buffer with 0.1% formaldehyde), light protected and incubated for 30 minutes at 4 °C. Cells were washed in FACS wash buffer with centrifugation then transferred to tubes with 500 μL FACS wash buffer. Flow cytometric analysis was performed on a BD LSRII (San Jose, CA). Antibodies used include: From BD Biosciences: 1:800 dilution of CD11cBV786 (cat # 563735), 1:50 dilution of CD45.2 BV605 (56305), 1:1000 dilution of CD3 PerCP (561089), and 1:800 dilution of Ly6G Fitc (551460). From eBiosciences: 1:4000 dilution of CD4 e450 (48-0041-80), 1:2000 dilution of CD8 APCe780 (47-0081-80), 1:400 dilution of NK1.1 APCe (17- 5941-63), 1:4000 dilution of CD11b PECy7 (15-0112-81), 1:4000 dilution of CD19 Pee610 (61-0193-80), 1:4000 dilution of Ly6C PE (12-5932-80) and 1:200 dilution of CD16/CD32 (16-0161-81). From Biolegend: 1:100 dilution of CX3Cr1 BV421 (SA011F11).

**Human Fis1 plasmid transfection and mitochondria staining**. HEK293T cells (CRL-11268, ATCC) were cultured in DMEM-GlutaMAX™ (Gibco) media containing 10% FBS (Gibco) with 1% antibiotic–antimycotic (Gibco). HEK293T cells were seeded on coverslips in 24-well plates at $5 × 10^4$ cells per well overnight. N-terminal His-tagged Human Fis1 expression plasmid on pCMV3 vector (His-hFis1)(HG14981-NH, Sino Biological) was transiently expressed in seeded HEK293T cells using Lipofectamine™ 2000 (11668019, Invitrogen) transfection reagent according to manufacturer's instructions (Invitrogen). Control cells received only pCMV3 vector plasmid (CV011, Sino Biological)[72]. Cells were treated with 100 μM METH (dissolved in saline) and saline (as vehicle control) post-24 hours of transfection. After 24 hours of exposure, cells were fixed and permeabilized with 4% paraformaldehyde with 0.5% v/v Triton-x 100 in PBS (pH 7.4) for 20 minutes at RT. Next, formaldehyde was quenched with 0.1 M glycine (pH 3.5) for 30 minutes at RT. The cells were blocked with 1% bovine serum albumin, 0.1% cold water fish skin gelatin, 0.1% Tween-20, and 0.05% sodium azide in PBS (pH 7.4) for 1 hour at RT. Cells were then immunolabeled with mouse Tom20 (1:100; ab56783, Abcam) to stain intracellular mitochondrial networks and goat anti-His Tag (1:100, A190-113A, Bethyl Laboratories) primary antibodies to detect exogenously expressed Fis1 protein followed by anti-mouse and anti-goat secondary antibodies with Alexa 488 and Alexa 568 dyes (Invitrogen), respectively, for 1 hour at RT. Next, cells were washed with PBS and mounted with Vectashield Hardset (Vector Laboratories) mounting media. Immunofluorescently labeled HEK293T cells were subsequently observed on Nikon A1R high-resolution confocal microscope (Nikon) and imaged with Nikon NIS Elements software (v4.13.04) with a ×60 oil objective lens (NA = 1.4). To quantify mitochondrial networks morphological parameters, images were first convolved using NIH ImageJ (v1.6.0) software's convolve filter to isolate and equalize fluorescent pixels as reported earlier[76]. Convolved images were then used to quantify mitochondrial lengths (μm) and widths (μm) using Nikon NIS Elements software (v4.13.04). The mitochondrial aspect ratio was calculated from the ratio of length and width. All image acquisition and analysis were performed in an analyst-blinded fashion.

**Statistics and reproducibility**. All cell culture experiments were repeated two to three times. For all in vivo studies, investigators were blinded with respect to the mouse treatment groups. We used a numerical ear tagging system for unbiased data collection. For all imaging studies, paraffin blocks, slides, and acquired microscopic images were alphanumerically labeled. Following the completion of the study, individual mouse ID and image ID numbers were cross-referenced with treatment to permit analysis. All statistical analyses were conducted in GraphPad Prism software (v8.2.1, La Jolla, CA). Data were presented in graphs showing median and interquartile ranges. Two-tailed, unpaired $t$ test (for two groups), one-way ANOVA (for three groups or more), two-way ANOVA (for multiple variables with repeated measures) were used, followed by Tukey's and Sidak's multiple

comparisons post hoc test. $P < 0.05$ (95% confidence interval) were considered significant.

**Reporting summary**. Further information on research design is available in the Nature Research Reporting Summary linked to this article.

## Data availability

Source data for all graphs in this article are included in Supplementary Data 1. Uncropped images for all western blots in this article are included in the Supplementary Information. Other information and data in this study are available from the corresponding author on reasonable request.

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

## Acknowledgements
This work was supported by the National Institutes of Health grants: HL122354 and HL145753 to M.S.B; LSUHSC-S CCDS Finish Line Award, COVID-19 Research Award, and Feist Weiller Cancer Center IDEA Grant to M.S.B; P20GM121307 and R01HL149264 to C.G.K; NIH R01 HL098435, HL133497, and HL141155 to A.W.O; HL141998 to S.M.; AA025744 and AA026708 to M.P.; HL131844 to M.D.W.; LSUHSC-S Malcolm Feist Cardiovascular and AHA Postdoctoral Fellowship to C.S.A. (20POST35210789); AHA Postdoctoral Fellowship to S.A. (18POST34080495); and LSUHSC-S Malcolm Feist Pre-doctoral Fellowship to R.A.

## Author contributions
C.S.A., R.A., and M.S.B. conceptualized the study; C.S.A., R.A., and M.S.B. designed the experiments; C.S.A., R.A., S.A., M.M., N.S.R, and S.N. performed all experiments and participated in analyses; C.S.A., R.A., and G.K. injected mice with METH; J.T. collected autopsy human hearts and toxicological reports; S.A. performed primary cardiomyocyte isolation; M.M. assisted in measuring the mitochondrial size distribution in cardiomyocytes; N.S.R. performed mRNA experiments and analysis; S.M. and M.P. performed the Seahorse experiments and data analysis while blinded to the animal and treatment IDs; S.C. and M.D.W. performed immunophenotyping experiments; B.H. and J.K. performed electron microscopic experiments; X.Y. performed IHC staining; M.A.N.B. performed statistical analysis; X.L., K.S. D.P., and C.L.A. contributed to analytic tools; A.W.O., C.G.K., and N.E.G. contributed to reagents; C.S.A. and M.S.B. wrote the manuscript, and all of the authors have read, edited, and approved the paper.

## Competing interests
The authors declare no competing interests.
