## [Peer Review File · Communications Biology]

Reviewers' comments:

Reviewer #1 (Remarks to the Author):

In this study, the authors assessed the direct role of Sigmar1 on methamphetamine (METH)-associated cardiomyopathy. Autopsy samples of human hearts were collected, and a 'binge and crash' METH administration mouse model was used. METH users' hearts also showed cardiomyopathy signs, including cellular injury, fibrosis, and enlargement of the heart. Mice exposed to 'binge and crash' METH develop cardiac hypertrophy, fibrotic remodeling, and mitochondrial dysfunction leading to contractile dysfunction. METH-treatment inhibits Sigmar1 resulting in inactivation of cAMP response element-binding protein (CREB), resulting in decreased expression of mitochondrial fission 1 protein (FIS1) that ultimately lead to alteration in mitochondrial dynamics and function. This study revealed a direct role of Sigmar1 as a viable therapeutic target in protecting against METH-associated cardiomyopathy.

Comments.

1. The authors found that METH group was characterized by a reduced cardiac systolic dysfunction not the diastolic function. The mechanism accounting for this diastolic function result should be better explained. The expression levels of fetal ANP, BNP and β -MHC accounting for this result should be added.

2. A Representative figures of FACS results in Supplementary Figure 1c and d are missing.

3. How did four week-'binge and crash' METH treatment model could influence mice heart Sigmar-1 and inactivation of cAMP response element-binding protein (CREB), resulting in decreased expression of mitochondrial fission 1 protein (FIS1). What is the result of immunoprecipitation results of Sigmar-1 and CREB. It's better to provide a graphic abstract.

Fig 7f, the DAPI in the Vehicle group is not clear enough.

Fig. 8d, describe this result in the 'result' part(compare Fig. 3G).

Described pattern on scale bar in different images is not consistent, for example, Fig. 7f without details but Fig. 9d with 20 μ m.

Reviewer #2 (Remarks to the Author):

METH abuse is associated with cardiovascular toxicity and cardiomyopathy, albeit with a less-understood mechanism. Studies by Abdullah et al. utilize multiple complementary approaches to address this knowledge gap. The authors investigated the pathological remodeling of the heart associated with METH illicit use in human autopsy hearts obtained from METH users and control group (non-METH users) based on toxicological reports (the 1st strength of this study). The cellular mechanism was examined in a preclinical 'binge and crash' METH administration mouse model. This animal model seems to recapitulate the pathological features observed in human METH-users' heart. Employing this animal model, the authors determined the molecular mechanisms of METH-induced cardiomyopathy (the 2nd strength of this study). The experimental design is logical; data analysis and presentation of the data are compelling. The idea that the sigma-1R agonist such as PRE-084, (or fluoxetine, an FDA approved drug and a potent sigma-1R

agonist) may serve as a therapeutic agent, is clinically relevant.

Addressing the concerns outlined below will enhance the impact of this report:

For mitochondrial respiration, a control group such as norepinephrine should be included. Inclusion of a time course for METH-induced mitochondrial oxidative respiration (i.e., progressive change after one, two, three, and four weeks of METH exposure) will strengthen the conclusions made.

For in vitro studies (respiration and neonatal cardiomyocytes), what is the justification for the use of 100 μ M METH?

While METH attenuation of CREB expression and activation in cardiomyocytes is one of the most exciting findings in this study, the mechanism has not been studied. A possible link might be METH regulation of intracellular calcium and thus CREB phosphorylation.

While the current study and previous publications by this group clearly show the genetic loss of the sigma-1R (global sigma1R^{-/-} mice) suppresses mitochondrial respiration, and hyper fused mitochondria in mouse heart, it is still unclear how METH decreases the sigma-1R levels in the cardiomyocytes. Therefore, this reviewer's main concern is: how the sigma1-R expression level is reduced in the cardiomyocytes of human METH-users and the mouse model of METH exposure used in this study?

Minor comments:

In supplemental Figure 1, only selected markers of innate and adaptive immune cells are tested (which is fine). Therefore, please revised the statement below to reduced the generality of the statement: "...we conclude that four weeks 'binge and crash' METH administration to mice did not alter major innate and adaptive immune cells population in spleen and systemic circulation....".

Please include the flow cytometry panels.

If data related to animals' weight, locomotor activity, and blood concentration of METH are available, please include them.

Reviewer #3 (Remarks to the Author):

Methamphetamine (METH) is one of the most abused drugs leading to substantial loss on both society and individuals. Cardiovascular system pathology is well-known in Meth users; however, the detailed molecular mechanisms remain much unexplored. The submitted manuscript focused on this interesting topic and identified the critical role of sigma 1 receptor during this process. Through human studies and animal models, the authors demonstrated that METH could impair cardiocyte mitochondrial function through decreasing FIS1 levels. At the molecular levels, METH decreased the levels of sigma 1 receptor, which in turn, resulted in reduced activity of CREB/FIS1 axis. These findings were further demonstrated by genetic approach using overexpressing sigma 1 receptor. In general, this manuscript targets an interesting topic based on solid research background with well-organized structure, provides compelling results by using multiple approaches. Overall, this manuscript is in good quality; however, there are some concerns needed to be addressed.

1) In the "Binge and Crash" experiments, what is the group size? The authors mentioned all mice are male, what is the rationale for this? Any gender differences?

2) The authors argued that FIS1 downregulation is critical for METH-mediated mitochondria in vitro and in vivo. This declaration should be further supported by using FIS1 overexpression plasmids.

3) The sigma 1 receptor is well-known to regulate mitochondrial function due to its specific location on the mitochondria-associated membrane of the endoplasmic reticulum. Previous studies have demonstrated that the translocation of sigma receptor is critical for cocaine-mediated biological effects. Can Meth also induce the similar translocation in cardiocytes?

4) The authors mentioned that in sigma 1 receptor KO mice we could observe the pathology on heart. The authors should also detect the expression levels of FIS1 and total and phosphorylation levels of CREB in these mice.

Suggestion: accept after revision.

Response to Reviewer #1

Q. In this study, the authors assessed the direct role of Sigmar1 on methamphetamine (METH)-associated cardiomyopathy. Autopsy samples of human hearts were collected, and a 'binge and crash' METH administration mouse model was used. METH users' hearts also showed cardiomyopathy signs, including cellular injury, fibrosis, and enlargement of the heart. Mice exposed to 'binge and crash' METH develop cardiac hypertrophy, fibrotic remodeling, and mitochondrial dysfunction leading to contractile dysfunction. METH-treatment inhibits Sigmar1 resulting in inactivation of cAMP response element-binding protein (CREB), resulting in decreased expression of mitochondrial fission 1 protein (FIS1) that ultimately lead to alteration in mitochondrial dynamics and function. This study revealed a direct role of Sigmar1 as a viable therapeutic target in protecting against METH-associated cardiomyopathy.

Response: We thank the reviewer for the thoughtful insights and comments. We addressed all the concerns with additional experiments and data in the revised manuscript.

Q1. The authors found that METH group was characterized by a reduced cardiac systolic dysfunction not the diastolic function. The mechanism accounting for this diastolic function result should be better explained. The expression levels of fetal ANP, BNP, and β -MHC accounting for this result should be added.

Response: Most studies on METH-associated cardiomyopathy showed the evidence of systolic cardiac dysfunction indicated by lower left ventricular (LV) ejection fraction (EF) measured by echocardiography. Accumulating clinical studies reported severe LV ejection fraction attenuation with LV dilatation in METH users indicating the inherent existence of METH use associated cardiomyopathy (Ito, H., et al. *Clin Cardiol.* 2009, 32: E18–E22). Studies on patients with a discharge diagnosis of cardiomyopathy and concomitant METH abuse revealed a high percentage (19 of 21 patients) of echocardiographic LV-dilatation LV EF reduction (Wijetunga, M., et al. *J Toxicol Clin Toxicol.* 2003, 41: 981–986).

In this manuscript, we didn't measure any diastolic functional parameters after METH treatment. Measurement of diastolic function in mice requires special techniques including Pulse wave Doppler and Tissue Doppler echocardiography as well as Presser-Volume loop measurement using invasive catheterization. Though these techniques are currently available in our facility, we cannot perform the additional experiment at this point. We aim to critically evaluate different forms of cardiac functional abnormalities in our future studies. Activation of the fetal gene program (ANP, BNP, and β -MHC) in the adult heart used as a biomarker of cardiac hypertrophy (Taegtmeyer, H., et al. *Ann N Y Acad Sci.* 2010, 1188:191-8) and is prominently augmented in the ventricles in severe congestive heart failure (Hasegawa, K., et al. *Circulation.* 1993, 88:372-380; Mukoyama, M., et al. *J Clin Invest.* 1991, 87:1402-1412). A recent study of 4407 patients positive for METH use showed that 714 patients were screened for heart failure, and 450 had abnormal levels of BNP (Richards, J.R., et al. *Am J Emerg Med.* 2018, 36: 1423–1428). The prevalence of abnormal BNP (> 100 pg/mL) in the METH-tested patient group was 10.2% (450/4407) versus 6.7% (7263/108,608) in the combined methamphetamine-negative and non-tested groups. Moreover, echocardiography studies showed a significant difference in normal LVEF (50–70%) and severely dysfunctional LVEF (<30%) for METH-positive patients with normal versus abnormal BNP. Logistic regression analysis revealed predictors of abnormal BNP and LVEF < 30% in METH-positive patients, including age, race, smoking history, elevated creatinine, and respiratory rate. In the present study, we observed significantly decreased LV EF and increased fetal gene program (ANP, BNP, and β -MHC) in METH-treated mice heart. As the reviewer suggested, we described this in the discussion of the revised manuscript.

Q2. A Representative figures of FACS results in Supplementary Figure 1c and d are missing.

Response: As the reviewer suggested, we included the flow cytometry panels in Supplement Figure 1a and 1c in the revised manuscript.

Q3. How did four week-'binge and crash' METH treatment model could influence mice heart Sigmar-1 and inactivation of cAMP response element-binding protein (CREB), resulting in decreased expression of mitochondrial fission 1 protein (FIS1). What is the result of the immunoprecipitation results of Sigmar-1 and CREB. It's better to provide a graphic abstract.

Response: We did not perform any co-immunoprecipitation experiment in this manuscript. Future studies are required to determine the precise molecular mechanisms of Sigmar1-CREB signaling pathways in cardiomyocytes mediated by METH exposure. We also didn't include a graphical abstract as Communications Biology does not allow us to add a graphical abstract.

Q4. Fig 7f, the DAPI in the Vehicle group is not clear enough.

Response: As the reviewer suggested, we improved the image quality to represent the DAPI in the vehicle group in the revised manuscript.

Q5. Fig. 8d, describe this result in the 'result' part (compare Fig. 3G).

Response: We describe the Fig. 8d in the result section of the manuscript.

It is very difficult to correlate the FCCP-stimulated maximal respiration in cardiac mitochondria (Fig. 8d) data with the unaffected heart rate in METH treated mice (Fig. 3g). The literature on the effect of METH on heart rate is contradictory even though all studies suggest a decrease in cardiac contractility. For instance, METH exposure has been shown to directly attenuate maximal velocity of left ventricular pressure development (+dP/dt) and decline (-dP/dt) without affecting heart rate (bpm, beats per minute) in isolated mouse hearts *ex vivo* at a concentration of 0.1 mM (100 μ M) and 1 mM (1000 μ M) for 30 minutes (Turdi, S., et al. *Toxicol Lett.* 2009, 189:152-158). In contrast, METH increases the beating rate at 500 μ M in isolated neonatal rat ventricular cardiomyocytes *in vitro* (Sugimoto, K., et al. *Biochem Biophys Res Commun.* 2009, 390:1214-1220). Moreover, 8-weeks of METH binge administration (binge period: 4 mg/kg twice daily for 4 days, followed by a 10-day drug-free period, total 4 binges) to rats have not shown any significant difference in heart rates (bpm) (Lord, K.C., et al. *Cardiovasc Res.* 2010, 87:111-118). Whereas, binge METH exposure from 15 mg/kg escalated to 40 mg/kg five days per week for three months shown a slight decrease in heart rates in mice compared to Saline only treatment (Yu, Q., et al. *Life Sci.* 2002, 71(8):953-965). These apparent discrepancies in heart rates (bpm) in METH exposed hearts in preclinical studies underscores how METH dose, dosing regimen, and exposure period determine the effect of METH on heart rates. In our current study, we found that 4-weeks of METH 'binge and crash' treatment protocol suppresses FCCP-stimulated maximal respiration in cardiac mitochondria (Fig. 8d). In contrast, we have not found any significant difference in heart rates between vehicle and METH treated mice (Fig. 3g). Therefore, we cannot explain these two events and correlate them as well based on our findings.

Q6. Described pattern on scale bar in different images is not consistent, for example, Fig. 7f without details but Fig. 9d with 20 μ m.

Response: As the reviewer suggested, we included the details about the scale bar in all the figures.

Response to Reviewer #2

Q. METH abuse is associated with cardiovascular toxicity and cardiomyopathy, albeit with a less-understood mechanism. Studies by Abdullah et al. utilize multiple complementary approaches to address this knowledge gap. The authors investigated the pathological remodeling of the heart associated with METH illicit use in human autopsy hearts obtained from METH users and control group (non-METH users) based on toxicological reports (the 1st strength of this study). The cellular mechanism was examined in a preclinical 'binge and crash' METH administration mouse model. This animal model seems to recapitulate the pathological features observed in human METH-users' heart. Employing this animal model, the authors determined the molecular mechanisms of METH-induced cardiomyopathy (the 2nd strength of this study). The experimental design is logical; data analysis and presentation of the data are compelling. The idea that the sigma-1R agonist such as PRE-084, (or fluoxetine, an FDA approved drug and a potent sigma-1R agonist) may serve as a therapeutic agent, is clinically relevant. Addressing the concerns outlined below will enhance the impact of this report.

Response: We thank the reviewer for the thoughtful insights and comments. We addressed all the concerns with additional experiments and data in the revised manuscript.

Q1. For mitochondrial respiration, a control group such as norepinephrine should be included.

Response: As the reviewer suggested, we treated the cardiomyocytes with norepinephrine and provided the data in the Supplement Figure 4 of the revised manuscript. Human METH users who experienced METH-induced psychosis have been shown to have norepinephrine (NE) level as high as 0.89 nM (0.89 pmol/mL) in their plasma (Yui, K., et al. *Ann N Y Acad Sci.* 2004, 1025:296-306). In contrast, 'binge and crash' METH administration has been shown to increase NE level in plasma to average 28.54 nM (4.83 ng/mL) concentration compared to Saline treated mice (6.80 nM, 1.15 ng/mL) (Lee, M. et al. *Exp Eye Res.* 2020, 193:107964). Hence, in our current study, we treated isolated cardiomyocytes with 5 nM, 25 nM, and 50 nM NE to measure mitochondrial respiratory profiles.

Q2. Inclusion of a time course for METH-induced mitochondrial oxidative respiration (i.e., progressive change after one, two, three, and four weeks of METH exposure) will strengthen the conclusions made.

Response: As the reviewer suggested, we performed additional experiments in mice treated with METH for 2 weeks and measured mitochondrial respiration. We provided the new data in the Supplement Fig of the revised manuscript.

Q3. For in vitro studies (respiration and neonatal cardiomyocytes), what is the justification for the use of 100µM METH?

Response: Human METH users exhibited METH concentrations averaging from 2 µM to 3 µM. In comparison, METH levels can reach 17 µM in individuals arrested for erratic behavior and 87 µM in postmortem samples from non-overdose patients (Melega, W. P., et al. *Synapse.* 2007, 61, 216-220; McIntyre, I. M., et al. *J Anal Toxicol.* 2013. 37, 386-389). In preclinical studies, both neonatal rat cardiomyocytes and adult murine cardiomyocytes have been exposed to METH at concentrations ranging from 100 µM to 500 µM to study METH-induced changes in

cardiomyocytes contractility (Sugimoto, K., et al. *Biochem Biophys Res Commun.* 2009, 390:1214-1220; Turdi, S., et al. *Toxicol Lett.* 2009, 189:152-158). Hence, to mimic long term exposure of METH to cardiomyocytes and based on our literature survey, we have used 100 μM of METH for our neonatal rat cardiomyocyte studies.

Q4. While METH attenuation of CREB expression and activation in cardiomyocytes is one of the most exciting findings in this study, the mechanism has not been studied. A possible link might be METH regulation of intracellular calcium and thus CREB phosphorylation.

Response: We agree with the reviewer that METH regulation of intracellular calcium may be responsible for CREB phosphorylation. Studies also demonstrated that type 1 and type 3 IP3 receptors were mainly located in the nuclear fraction in cardiomyocytes and myotubes (Ibarra, C., et al. *J Biol Chem.* 2004, 279:7554-7565; Cardenas, C., et al. *J Cell Sci.* 2005; 118: 3131-3140). Studies showed that IGF-1 induced a fast and transient increase in intracellular Ca^{2+} levels apparent both in the nucleus and cytosol, releasing this ion from intracellular stores through an inositol 1,4,5-trisphosphate (IP3)-dependent signaling pathway (Ibarra, C., et al. *J Biol Chem.* 2004, 279:7554-7565). Ca^{2+} signals mediated by nuclear IP3 receptors in muscle causes a rapid CREB phosphorylation (Cardenas, C., et al. *J Cell Sci.* 2005; 118: 3131-3140). Extensive studies suggested that Sigmar1 functions to regulate the intracellular Ca^{2+} dynamics by modulating ion channels, including IP3 receptors (Maurice, T., et al. *Pharmacol Ther.* 2009, 124(2):195–206). Though we speculate that similar mechanisms are contributing to our findings, we didn't perform any experiment to prove this hypothesis and therefore, we didn't mention about this pathway in this manuscript. Future work will include a thorough interrogation of the pathway involved in our model.

Q5. While the current study and previous publications by this group clearly show the genetic loss of the sigma-1R (global sigma1R^{-/-} mice) suppresses mitochondrial respiration, and hyper fused mitochondria in mouse heart, it is still unclear how METH decreases the sigma-1R levels in the cardiomyocytes. Therefore, this reviewer's main concern is: how the sigma1-R expression level is reduced in the cardiomyocytes of human METH-users and the mouse model of METH exposure used in this study?

Response: CREB nuclear translocation may be an early event activated by METH as observed immediately after METH (30 mg/kg, 30 min) treatment in mice and in vitro METH (150 μM) exposure in primary rat astrocytes. Moreover, Sigmar1 regulates the CREB translocated into the nucleus as pretreatment with the Sigmar1 antagonist in astrocytes blocked CREB nuclear translocation. Sigmar1 gene has a CREB-binding site at its promoter region. METH-mediated activation of astrocytes involved CREB translocated into the nucleus, interaction with the promoter of Sigmar1, resulting in increased expression of Sigmar1. Similarly, Sigmar1 upregulation was observed in cultured astrocytes from adult rats after infection with the VP16-CREB viral vector, and in transgenic mice with targeted activation of CREB in astrocytes. All these studies suggest a CREB-Sigmar1 feedback loop regulating their expression modulated by the METH exposure. Future studies are required to determine the precise molecular mechanisms of Sigmar1 downregulation in cardiomyocytes mediated by METH exposure.

Minor comments:

Q6. In supplemental Figure 1, only selected markers of innate and adaptive immune cells are tested (which is fine). Therefore, please revised the statement below to reduced the generality of the statement: “.....we conclude that four weeks ‘binge and crash’ METH administration to mice did not alter major innate and adaptive immune cells population in spleen and systemic circulation....”.

Response: As the reviewer suggested, we corrected these in the revised manuscript.

Q7. Please include the flow cytometry panels.

Response: As the reviewer suggested, we included the flow cytometry panels in the Supplement Figure 1a and 1c in the revised manuscript.

Q8. If data related to animals' weight, locomotor activity, and blood concentration of METH are available, please include them.

Response: As the reviewer suggested, we included the animals weight data in Fig. 4d in the revised manuscript. We didn't perform the locomotor activity as the "Binge and Crash" model used in this study already reported the locomotor activity (Kesby, J.P., et al. *Addict Biol.* 2017, 23:206-218). We did not measure the blood concentration of METH.

Response to Reviewer #3

Q. Methamphetamine (METH) is one of the most abused drugs leading to substantial loss on both society and individuals. Cardiovascular system pathology is well-known in Meth users; however, the detailed molecular mechanisms remain much unexplored. The submitted manuscript focused on this interesting topic and identified the critical role of sigma 1 receptor during this process. Through human studies and animal models, the authors demonstrated that METH could impair cardiocyte mitochondrial function through decreasing FIS1 levels. At the molecular levels, METH decreased the levels of sigma 1 receptor, which in turn, resulted in reduced activity of CREB/FIS1 axis. These findings were further demonstrated by genetic approach using overexpressing sigma 1 receptor. In general, this manuscript targets interesting topics based on solid research background with a well-organized structure that provides compelling results by using multiple approaches. Overall, this manuscript is in good quality; however, there are some concerns needed to be addressed.

Response: We thank the reviewer for the thoughtful insights and comments. We addressed all the concerns with additional experiments and data in the revised manuscript.

Q1. In the "Binge and Crash" experiments, what is the group size? The authors mentioned all mice are male, what is the rationale for this? Any gender differences?

Response: As the reviewer suggested, we included the group size in the revised manuscript. Recent studies showed structural and molecular differences between female and male mice treated with methamphetamine 5 to 40 mg/kg over 5 months (Marcinko, M. C., et al. *Physiol Rep.* 2019, 7(6): e14036). Therefore, we used only male mice to avoid the effects of gender biases to our study.

Q2. The authors argued that FIS1 downregulation is critical for METH-mediated mitochondria in vitro and in vivo. This declaration should be further supported by using FIS1 overexpression plasmids.

Response: Cardiomyocytes are very difficult to plasmid transfection and require adenoviral infection to overexpress a protein. Due to time constraints and COVID-19 pandemic, it is beyond ability to generate FIS adenovirus to perform these experiments using cardiomyocytes. However, to address the reviewer's suggestion, we used HEK293 cells, which is easy to

perform plasmid transfection. As the reviewer suggested, we performed additional experiments using human FIS1 overexpression plasmids in HEK293T cells. We included the data in Supplement Figure 5 in the revised manuscript.

Q3. The sigma 1 receptor is well-known to regulate mitochondrial function due to its specific location on the mitochondria-associated membrane of the endoplasmic reticulum. Previous studies have demonstrated that the translocation of sigma receptor is critical for cocaine-mediated biological effects. Can Meth also induce the similar translocation in cardiocytes?

Response: Despite extensive studies over the last 40 years, Sigmar1's subcellular localization remains obscure. Sigmar1 was reported to localize at mitochondrial-associated membranes (MAM) (co-localized with Mito-DsRed) in CHO Cells and the plasma membrane, where it interacts with ion channels (Hayashi, T., et al. *Cell*. 2007, 131:596-610). Immuno-electron microscopic (EM) data from the Mavlyutov group showed Sigmar1's sub-cellular localization largely depends on cell and organ types. For example, Sigmar1 localizes to nuclear envelope in photoreceptor cells where no localization was observed in ER, nucleoplasmic reticulum, as well as in the nucleus in NSC34 cell line (Mavlyutov, T. A., et al. *Sci Rep*. 2015, 5:10689, Mavlyutov, T. A., et al. *Oncotarget*. 2017, 8:51317-51330). Moreover, Immuno-EM examinations were unable to detect Sigmar1 at the plasma membrane. Interestingly, subcellular fractionation of neural tissues from the mutant SOD1 Tg mice showed Sigmar1 accumulation in mitochondrial fractions (Watanabe, S., et al. *EMBO Mol Med*. 2016, 8(12):1421-1437). A recent study from Mavlyutov et al. showed Sigmar1's C-terminal resides inside ER-lumen, and the N-terminus resides in the cytosol, which is opposite to the recently derived crystal structure proposing that Sigmar1's C-terminal reside on the cytosolic side of the ER membrane (Mavlyutov, T. A., et al. *Protein & Cell*. 2018, 9:733-737, Schmidt, H.R., et al. *Nature*. 2016, 532:527-530). These studies suggest organ-specific subcellular localization and function of Sigmar1. However, Sigmar1's subcellular distribution in pathophysiological conditions in the heart remains unknown. Therefore, we don't know whether Sigmar1 translocation is affected by METH, and our future studies will aim to determine whether METH affects Sigmar1 translocation to subcellular organelle and elicit its effect in cardiomyocytes.

Q4. The authors mentioned that in sigma 1 receptor KO mice we could observe the pathology on heart. The authors should also detect the expression levels of FIS1 and total and phosphorylation levels of CREB in these mice.

Response: As the reviewer mentioned, our recent study suggested Sigmar1 loss of function led to mitochondrial dysfunction, abnormal mitochondrial architecture, and adverse cardiac remodeling, culminating in cardiac contractile dysfunction (Abdullah, C.S., et al. *J Am Heart Assoc*. 2018, 7(20): e009775). Due to the COVID-19 pandemic associated shutdown of the institution and limited access to animals and resources, we plan to perform future detailed experiments to determine the Sigmar1 dependent regulation of CREB-FIS1 signaling using Sigmar1 overexpression and knockout mice models.

REVIEWERS' COMMENTS:

Reviewer #2 (Remarks to the Author):

This is an exciting and timely study. The use of complementary approaches and inclusion of multiple control groups have increased the rigor and reproducibility of this work. The authors have addressed this reviewers concern.